# DSTYK phosphorylates STING at late endosomes to promote STING signaling

Hao Dong [iD][1,3✉], Heng Zhang [iD][1,3], Pu Song[1], Yuan Hu[2] & Danying Chen [iD][1✉]

## Abstract

Stimulator of interferon genes (STING) is essential for innate immune pathway activation in response to pathogenic DNA. Proper activation of STING signaling requires STING translocation and phosphorylation. Here, we show that dual serine/threonine and tyrosine protein kinase (DSTYK) directly phosphorylates STING Ser366 at late endosomes to promote the activation of STING signaling. We find that TBK1 promotes STING post-Golgi trafficking via its kinase activity, thereby enabling the interaction between DSTYK and STING. We also demonstrate that DSTYK and TBK1 can both promote STING phosphorylation at late endosomes. Using an in vivo *Dstyk*-knockout model, we showed that mice deficient in DSTYK demonstrate reduced STING signaling activation and are more susceptible to infection with a DNA virus. Together, we reveal the previously unknown cellular function of DSTYK in phosphorylating STING and our findings provide insights into the mechanism of STING signaling activation at late endosomes.

**Keywords** DSTYK; Innate Immunity; Phosphorylation; STING; TBK1
**Subject Categories** Membranes & Trafficking; Microbiology, Virology & Host Pathogen Interaction; Signal Transduction

## Introduction

Cellular recognition and defense systems against microbial pathogens are fundamental features of higher organisms. The first line of defense in mammals is orchestrated by the innate immune system. Germline-encoded pattern recognition receptors detect various pathogen- and damage-associated molecular patterns, initiating signaling cascades that lead to the production of interferons (IFNs), IFN-stimulated genes (ISGs) and proinflammatory cytokines (Akira et al, 2006).

The cGAS-STING pathway is intricately involved in the rapid detection of aberrant DNA (as a pathogen-associated molecular pattern) and initiation of the innate immune response. The cytosolic DNA sensor cyclic guanosine monophosphate–adenosine monophosphate (cGAMP) synthase (cGAS) detects and binds pathogenic, genomic or mitochondrial DNA that is aberrantly

localized in the cytoplasm (Sun et al, 2013; Wu et al, 2013) and catalyzes the synthesis of the second messenger 2′3′-cGAMP (Ablasser et al, 2013; Diner et al, 2013; Gao et al, 2013), which binds and activates the endoplasmic reticulum (ER) membrane protein STING (also known as MITA, ERIS or MPYS) (Ishikawa and Barber, 2008; Jin et al, 2008; Sun et al, 2009; Zhong et al, 2008). After binding cGAMP, STING aggregates and translocates from the ER membrane to the Golgi apparatus (Dobbs et al, 2015; Ishikawa and Barber, 2008; Shang et al, 2019), followed by post-Golgi trafficking to late endosomes, and finally to lysosomes where it is degraded (Gonugunta et al, 2017; Gui et al, 2019). Activated STING recruits and activates TANK-binding kinase 1 (TBK1) and, subsequently, the transcription factor IFN regulatory factor 3 (IRF3), leading to the production of type I IFNs (Liu et al, 2015; Tanaka and Chen, 2012; Zhang et al, 2019; Zhao et al, 2019). Activation of the cGAS–STING pathway also leads to the expression of genes encoding proinflammatory cytokines through phosphorylation and activation of NF-κB.

STING translocation from the ER to the Golgi has been widely demonstrated to be essential for the activation of STING signaling (Dobbs et al, 2015; Mukai et al, 2016; Ogawa et al, 2018), and recent reports have indicated that late endosomes are important organelles where STING can get fully activated (Fang et al, 2023; Gui et al, 2019). STING signaling is tightly regulated by post-translational modifications, such as phosphorylation (Hu et al, 2019; Li et al, 2015; Liu et al, 2015; Xia et al, 2019). Phosphorylation of STING at Ser366 plays an important role in STING signaling via recruitment of IRF3, thereby initiating the expression of type I IFNs (Liu et al, 2015; Tanaka and Chen, 2012). TBK1 is the main kinase that phosphorylates STING Ser366 to activate downstream signaling (Liu et al, 2015; Tanaka and Chen, 2012), but whether there are other kinases that can also promote the activation of STING signaling by phosphorylating STING Ser366 is still unknown.

DSTYK (also known as RIPK5) belongs to the receptor interacting protein kinase (RIPK) family due to its kinase domain homology (Zha et al, 2004) and is predicted to be a dual serine/threonine and tyrosine kinase (Peng et al, 2006). DSTYK induces both caspase-dependent and caspase-independent cell death when overexpressed (Zha et al, 2004), and studies in zebrafish have demonstrated the contributions of DSTYK toward development (Sun et al, 2020). Mutations in *DSTYK* are associated with congenital abnormalities of the kidney and urinary tract and spastic paraparesis type 23, resulting in abnormal development of the urinary tract and kidneys and paralysis, respectively (Lee et al, 2017; Sanna-Cherchi et al, 2013). Additional research has demonstrated the

[1]Key Laboratory of Cell Proliferation and Differentiation of the Ministry of Education, School of Life Sciences, Peking University, Beijing, China. [2]School of Life Sciences, Peking University, Beijing, China. [3]These authors contributed equally: Hao Dong, Heng Zhang. ✉E-mail: donghao98@stu.pku.edu.cn; dychen@pku.edu.cn

contributions of DSTYK toward cancer. Specifically, DSTYK promotes both TGF-β-induced epithelial–mesenchymal transition and subsequent chemoresistance in colorectal cancer cells (Zhang et al, 2020). Other RIPK family members are involved in the regulation of multiple innate immune pathways (Cuny and Degterev, 2021; He and Wang, 2018; Humphries et al, 2015; Silke et al, 2015); however, no connection between DSTYK and innate immunity have been reported so far, and potential substrates of DSTYK have yet to be identified.

Here, we show that DSTYK is a positive regulator of STING signaling. Specifically, DSTYK directly phosphorylates STING Ser366 at late endosomes to promote STING signaling. Additionally, TBK1 regulates DSTYK's role in controlling STING signaling activation through its kinase activity. And this regulation is achieved by managing the translocation of STING to late endosomes, thereby modulating its interaction with DSTYK. Besides, we also demonstrate that DSTYK and TBK1 can both phosphorylate STING Ser366 at late endosomes. Together, our findings suggest that DSTYK phosphorylates STING Ser366 at late endosomes. These results provide an understanding of the activation of STING signaling at late endosomes and may also lead to new treatments for STING-related immune diseases.

# Results

## DSTYK positively regulates the cytosolic DNA-triggered cGAS–STING pathway

We first analyzed the expression profile of DSTYK (Data Ref: Heng and Painter, 2008; Data Ref: Uhlen et al, 2010), and found that DSTYK is widely expressed among primary innate immune cells and human cell lines (Fig. EV1).

We silenced the expression of DSTYK in THP1-derived macrophages by short hairpin RNA (shRNA) in order to verify the function of DSTYK in innate immunity and tested a range of different stimuli for innate immune receptors (Fig. EV2A–E). And the results showed that DSTYK knockdown did not affect the immune pathway activation induced by Pam2CSK4 (TLR2 ligand), LPS (TLR4 ligand), poly (I:C) (TLR3 ligand) or Sendai virus (SeV; an RNA virus) (Fig. EV2A–D), but only inhibited herpes simplex virus 1 (HSV-1; a DNA virus) induced p-TBK1, p-IRF3, p-P65 and STING S366Pi (Fig. EV2E), which are recognized as markers for the activation of cGAS-STING pathway, and this indicates that DSTYK regulates the activation of cGAS-STING pathway.

We next silenced the expression of DSTYK in non-immune HeLa cells and HT1080 cells by shRNA to confirm its role in the activation of cGAS-STING pathway. Plasmid DNA (pDNA) transfection-induced production of IFNB, C-X-C motif chemokine ligand 10 (CXCL10; an ISG), interferon-induced protein with tetratricopeptide repeats 1 (IFIT1; an ISG) and the proinflammatory cytokine interleukin-6 (IL6) was inhibited in DSTYK-knockdown cells (Fig. EV2F,H), whereas responses triggered by infection with SeV were not altered (Fig. EV2G,I). DSTYK knockdown also inhibited the phosphorylation of TBK1, IRF3, STING, P65 and IκBα following pDNA transfection and infection with vaccinia virus (VACV; a DNA virus; Fig. EV3A,B,D,E,G); however, SeV-induced responses were not altered (Fig. EV3C,F,H).

To further confirm the role of DSTYK in activation of the cGAS–STING pathway, we deleted DSTYK in HeLa cells by using CRISPR–Cas9 (Fig. EV3I). Consistent with the results obtained using shRNA-mediated knockdown of DSTYK, pDNA transfection- and VACV-induced production of IFNB, CXCL10, IFIT1 and IL6 as well as phosphorylation of TBK1, IRF3, STING, P65 and IκBα were blunted in DSTYK-knockout cells generated using CRISPR–Cas9 (Figs. 1A,C,D and EV3J). Again, SeV-induced immune responses were not altered (Figs. 1B,E and EV3K). Together, these results suggest that DSTYK positively regulates activation of the cGAS–STING pathway triggered by cytosolic DNA.

## DSTYK interacts with STING at late endosomes

To investigate how DSTYK affects the activation of the cGAS–STING pathway, we first sought to determine the target of DSTYK. DSTYK-deficient cells showed impaired phosphorylation of TBK1, IRF3, and STING in response to cGAMP (STING ligand) stimulation (Fig. 2A,B). When examining the different interacting partners of DSTYK, we found that DSTYK specifically interacted with STING, but not with cGAS, TBK1 or RIG-I and VISA (the receptor and adapter involved in innate immune pathway in response to pathogenic RNA; Fig. 2C). Endogenous interaction between STING and DSTYK was also observed after pDNA stimulation (Fig. 2D). These results above indicate that DSTYK targets STING.

We next stably expressed Flag-tagged DSTYK in DSTYK-knockout HeLa cells and explored DSTYK subcellular localization during STING activation by immunofluorescence microscopy. Zebrafish DSTYK colocalizes with the late endosome markers Rab7 and LAMP1 when overexpressed in HEK293T cells (Sun et al, 2020). In agreement with these findings, stable expression of human DSTYK in HeLa cells showed colocalization of DSTYK with Rab7 and LAMP1, both at rest and following pDNA stimulation (Fig. 2E,F), but DSTYK did not colocalize with calnexin (ER marker), GRASP65 (Golgi marker) or EEA1 (early endosome marker; Fig. EV4A–C). Staining of DSTYK demonstrated a vesicular pattern that closely overlapped with Rab7 and LAMP1 (Fig. 2G,H), indicating its late endosomal membrane localization. These results confirm that DSTYK localizes at late endosomal membrane before or after STING activation.

To explore when STING translocates to late endosomes, we then tracked STING translocation after pDNA stimulation and observed that STING colocalized with GRASP65$^+$ Golgi for up to 3 h of pDNA stimulation, and after 3 h of stimulation, STING colocalization with GRASP65 weakened (Figs. 2I and EV4D). STING did not colocalize with Rab7$^+$ and LAMP1$^+$ late endosomes at 3 h of pDNA stimulation; however, at 8 h of pDNA stimulation, colocalization between STING and Rab7/LAMP1 was observed and continued to increase with increasing pDNA stimulation time (Figs. 2I and EV4E, F). Additional experiments using VACV stimulation demonstrated similar results (Fig. EV4G–I).

We next tracked the colocalization of STING and DSTYK with LAMP1 and Rab7. We found that STING clustered but did not colocalize with DSTYK and LAMP1/Rab7 after 3 h of stimulation (Fig. 2J,K), at which time STING colocalized with the Golgi apparatus (Fig. EV4D). STING, DSTYK, and LAMP1/Rab7 showed colocalization at 8 h of stimulation, and this colocalization continued to increase with increasing pDNA stimulation time (Fig. 2J,K). Based on the above, we conclude that DSTYK interacts with STING at late endosomes after stimulation of cytosolic DNA.

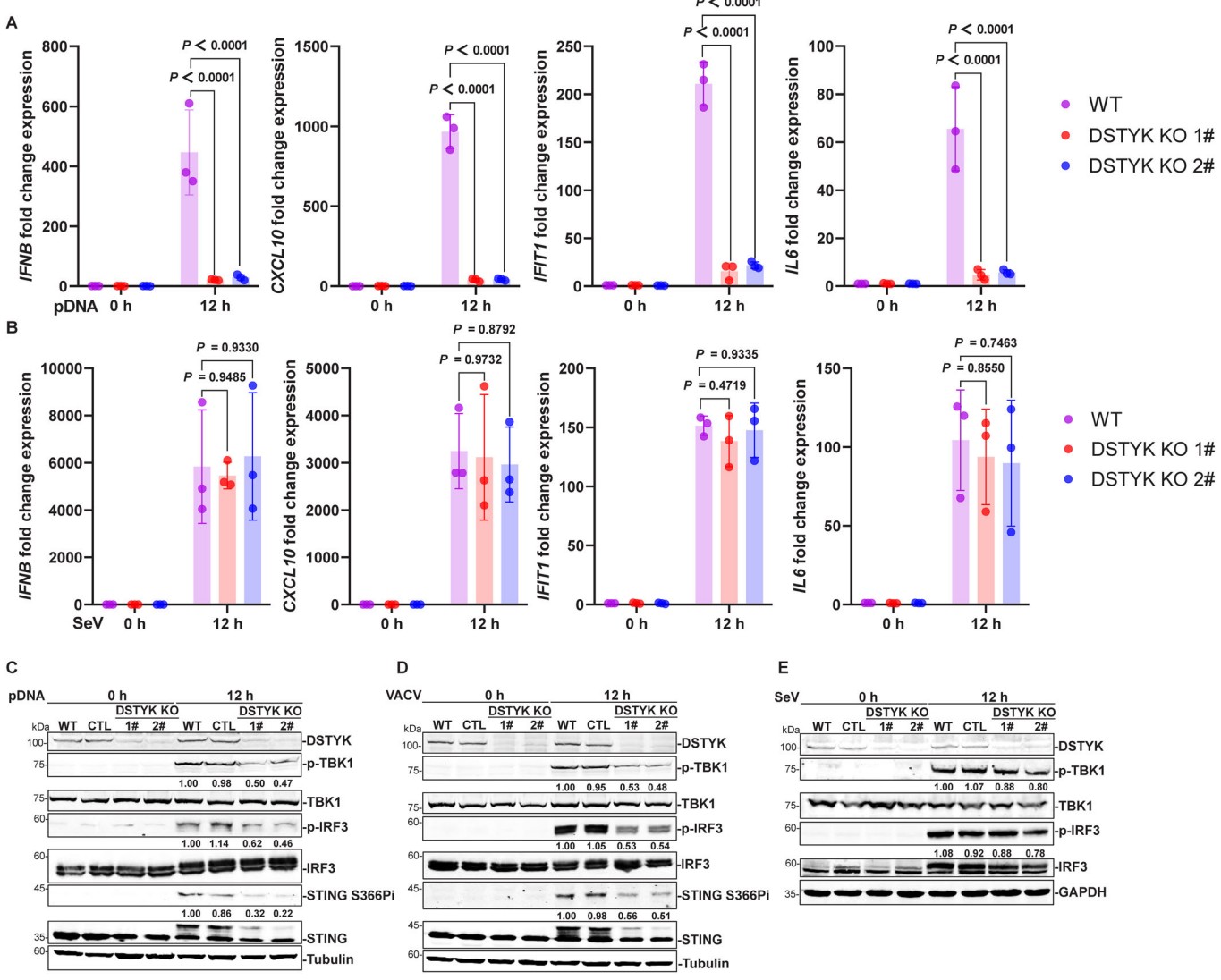

**Figure 1. DSTYK knockout inhibits the cytosolic DNA-triggered cGAS–STING pathway.**

(A, B) *DSTYK* knockout (KO) inhibits cytosolic DNA-triggered production of *IFNB*, ISGs and genes encoding proinflammatory cytokines. *DSTYK*-knockout HeLa cells and control cells were stimulated with pDNA transfection (1 μg/mL; **A**) and SeV (**B**) for 12 h, and the expression of detected *IFNB*, ISGs (*CXCL10* and *IFIT1*) and proinflammatory cytokines (*IL6*) was detected by quantitative PCR (*n* = 3 biological replicates). (C–E) *DSTYK* knockout inhibits cytosolic DNA-triggered phosphorylation of TBK1, IRF3, and STING at Ser366. *DSTYK*-knockout HeLa cells and control cells (WT and CTL) were stimulated with pDNA transfection (1 μg/mL; **C**), VACV (**D**) and SeV (**E**) for 12 h, and phosphorylation of TBK1, IRF3 and STING at Ser366 was detected by western blotting. Quantification of the indicated band intensities was performed in ImageJ, and quantification results are labeled below the indicated bands (*n* = 3 technical replicates). Data are representative of at least three independent experiments. Data in (**A**, **B**) are shown as the mean ± SD. *P* values were determined by two-way ANOVA and the exact *P* values are shown in the figures. Source data are available online for this figure.

## DSTYK directly phosphorylates STING Ser366 at late endosomes

We next wanted to narrow down the specific stage at which DSTYK regulates the activation of STING signaling. And we found that DSTYK deficiency did not affect the protein level of STING or STING colocalization with LAMP1 (Figs. 1C,D, 2A,B and EV3A,B,D,E, 5A), confirming that DSTYK does not affect STING degradation or translocation to late endosomes.

Since DSTYK is predicted to be a potential kinase, we wondered whether DSTYK regulates the activation of STING signaling via its

kinase activity and found that reconstitution of *DSTYK*-knockout cells with DSTYK kinase-dead mutants (DSTYK-K681A and DSTYK-D777A) did not restore the impaired pDNA transfection- and cGAMP-induced production of *IFNB* and ISGs as well as phosphorylation of IRF3 and STING Ser366 (Fig. 3A–C). However, DSTYK kinase-dead mutants still colocalized with LAMP1, and loss of DSTYK kinase activity did not inhibit DSTYK colocalization and interaction with STING (Fig. EV5B–D). These data demonstrate that the kinase activity of DSTYK is required for STING activation but does not affect its localization and interaction with STING.

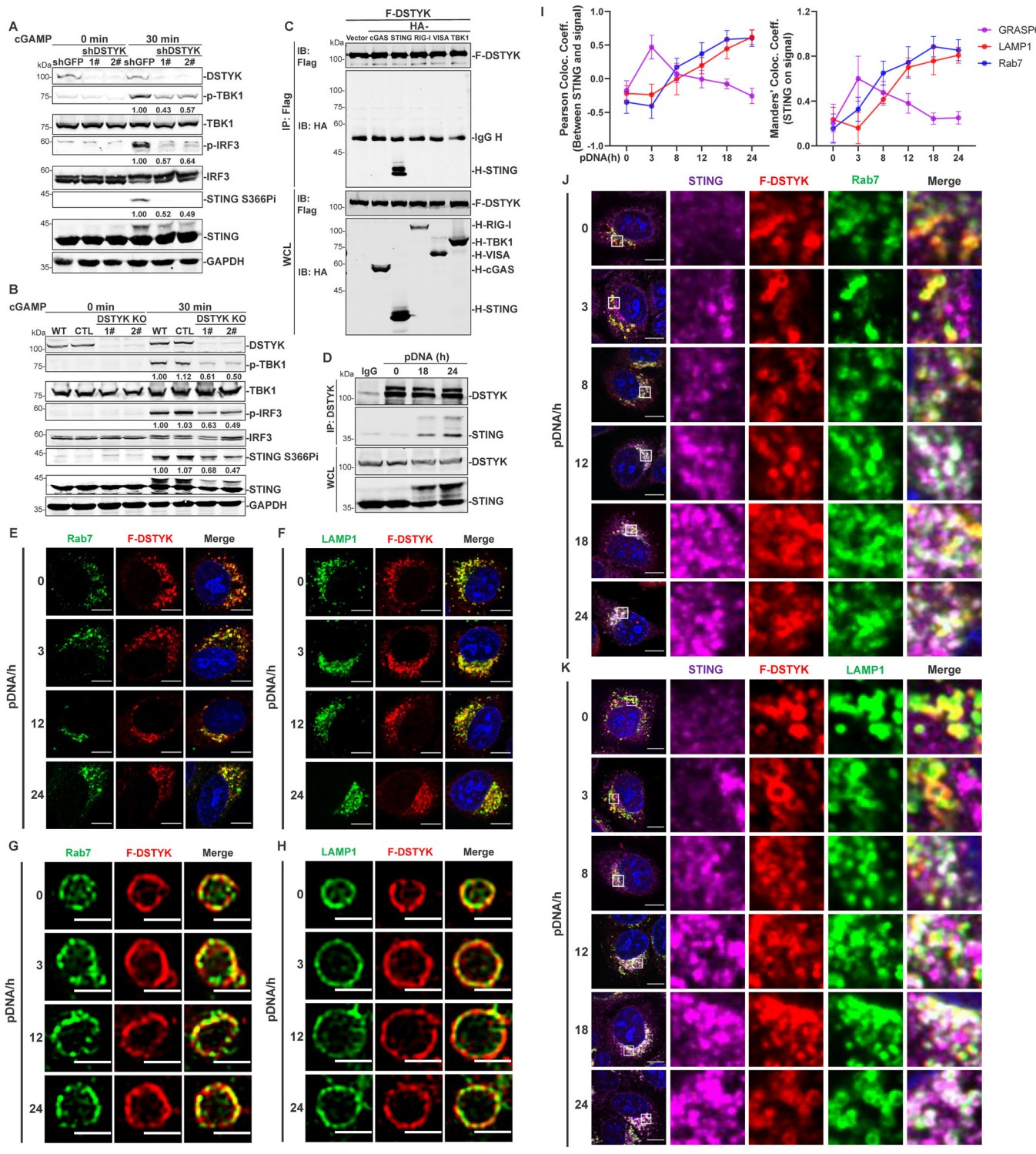

We therefore sought to explore whether DSTYK could phosphorylate STING via its kinase activity. Overexpression of STING with wild-type (WT) DSTYK, but not DSTYK kinase-dead mutants or truncations, in HEK293T cells caused STING phosphorylated band shifts (Figs. 2C, 3D). Together with our previous results showing that DSTYK deficiency impairs endogenous STING Ser366 phosphorylation (Figs. 1C,D, 2A,B and EV2E, 3A,B,D,E), we wondered whether

DSTYK phosphorylates STING at Ser366. Immunoblotting using an antibody to phosphorylated STING at Ser366 (S366Pi) and lysates from cells overexpressing both DSTYK and STING further confirmed that DSTYK could phosphorylate STING at Ser366 (TBK1 was used as a positive control; Fig. 3D). We next purified recombinant STING (amino acids 153–379) and incubated purified STING protein with WT DSTYK and kinase-dead mutants to perform in vitro kinase

**Figure 2. DSTYK interacts with STING at late endosomes.**

(A, B) DSTYK deficiency inhibits cGAMP-induced phosphorylation of TBK1, IRF3, and STING at Ser366. *DSTYK*-knockdown (A) and *DSTYK*-knockout (B) HeLa cells and control cells were stimulated with cGAMP (100 nM) for 30 min, and phosphorylation of TBK1, IRF3, and STING at Ser366 was detected by western blotting. Quantification of the indicated band intensities was performed in ImageJ, and quantification results are labeled below the indicated bands (n = 3 technical replicates). (C) DSTYK interacts with STING. STING-non-expressing HEK293T cells were transfected with the indicated Flag-tag (F) and HA-tag (H) plasmids (5 μg each) for 24 h. Whole-cell lysates (WCL) were examined, and cell lysates were subjected to immunoprecipitation (IP) with anti-Flag, followed by immunoblotting (IB) with anti-HA and anti-Flag. (D) Endogenous DSTYK interacts with STING after pDNA stimulation. HeLa cells were stimulated with pDNA (1 μg/mL) for 18 and 24 h. Whole-cell lysates (WCL) were examined, and cell lysates were subjected to immunoprecipitation (IP) with anti-DSTYK, followed by immunoblotting (IB) with anti-DSTYK and anti-STING. (E, F) DSTYK colocalizes with LAMP1 and Rab7 both at rest and after pDNA stimulation. *DSTYK*-knockout HeLa cells stably expressing DSTYK were stimulated with pDNA transfection (1 μg/mL) for the indicated amounts of time, followed by confocal imaging to observe DSTYK colocalization with Rab7 (E) and LAMP1 (F); scale bars, 10 μm. (G, H) DSTYK localizes at late endosomal membrane at rest and after pDNA stimulation. *DSTYK*-knockout HeLa cells stably expressing DSTYK were stimulated with pDNA transfection (1 μg/mL) for the indicated amounts of time, followed by super-resolution imaging to observe DSTYK colocalization with Rab7 (G) and LAMP1 (H); scale bars, 1 μm. (I) Quantification of the colocalization coefficients between STING and different organelles in Figure EV4D–F. Quantification of the colocalization coefficients was performed in ImageJ (n = 10 biological replicates). (J, K) Tracking the colocalization of STING, DSTYK and LAMP1/Rab7. *DSTYK*-knockout HeLa cells stably expressing DSTYK were stimulated with pDNA transfection (1 μg/mL) for the indicated amounts of time, followed by confocal imaging to observe STING and DSTYK colocalization with Rab7 (J) and LAMP1 (K); scale bars, 10 μm. Data are representative of at least three independent experiments. Data in (I) are shown as the mean ± SD. Source data are available online for this figure.

assays (Fig. 3E; TBK1 and kinase-dead mutant TBK1 K38A were used as controls). We observed that WT DSTYK, but not kinase-dead DSTYK mutants, directly phosphorylated STING at Ser366, which could be abolished by treatment with λ-phosphatase.

Considering the localization of DSTYK at late endosomes, we wondered whether DSTYK could phosphorylate STING at late endosomes. We first labeled phosphorylated STING Ser366 and directly tracked its intracellular trafficking to explore when phosphorylated STING localizes at late endosomes (Fig. EV5E–G). Phosphorylated STING colocalized with GRASP65 after 3 h of pDNA stimulation (Figs. 3F and EV5E). With the prolongation of pDNA stimulation time, phosphorylated STING colocalization with GRASP65 decreased, but colocalization with Rab7 and LAMP1 increased (Figs. 3F and EV5E–G). Based on these, immunofluorescence microscopy further tracked that phosphorylated STING, DSTYK, and LAMP1 showed colocalization at 8 h of stimulation, and the colocalization of these three increased with prolonged stimulation time (Fig. 3G).

Based on the above, we conclude that DSTYK directly phosphorylates STING Ser366 at late endosomes to promote STING signaling.

## TBK1 regulates the function of DSTYK to STING via controlling STING post-Golgi trafficking

TBK1 is currently recognized as the main kinase that phosphorylates STING at Ser366 (Liu et al, 2015). Activated STING uses its C-terminal TBK1-binding motif to recruit TBK1, which is activated by *trans*-autophosphorylation (Zhang et al, 2019; Zhao et al, 2019). Activated TBK1 then phosphorylates Ser366 in the pLxIS motif of the STING C terminus, and STING recruits IRF3 through phosphorylated Ser366 (Liu et al, 2015; Tanaka and Chen, 2012). Here, we identified DSTYK as an additional kinase that can also phosphorylate STING at Ser366. Based on these findings, we wanted to explore the relationship between DSTYK and TBK1 in regulating STING signaling further.

We generated *TBK1*-knockout HeLa cells and *TBK1*- and *DSTYK*-double-knockout (DKO) HeLa cells (Fig. 4A,B). In contrast to the downregulation of STING signaling in *DSTYK*-knockout cells, phosphorylation of IRF3 and STING at Ser366 as well as the

production of *IFNB* were completely abolished in both *TBK1*-knockout and *TBK1*- and *DSTYK*-DKO cells (Fig. 4A,B). Reconstitution with DSTYK in DKO cells did not restore the impaired pDNA transfection- and cGAMP-induced phosphorylation of IRF3 and STING Ser366 or the production of *IFNB* (Fig. 4C,D). Chemical inhibition of TBK1 kinase activity by treatment with BX795 inhibited the upregulation in IRF3 and STING Ser366 phosphorylation induced by cGAMP after reintroducing DSTYK expression in *DSTYK*-knockout cells (Fig. 4E). All these results suggest that DSTYK is dependent on TBK1 kinase activity to promote STING signaling.

We next explored how TBK1 kinase activity affects DSTYK function. TBK1 kinase activity has been shown to regulate STING post-Golgi trafficking in order to control STING degradation (Liu et al, 2022). We wondered whether TBK1 could direct the post-Golgi localization of STING and the interaction between STING and DSTYK, thereby influencing DSTYK's role in regulating STING signaling.

We first examined whether TBK1 affected STING trafficking via immunofluorescence microscopy. And we found that *TBK1* knockout did not affect STING colocalization with GRASP65 at 3 h of pDNA stimulation (Figs. 4F and EV6A), which indicates that TBK1 does not affect STING ER-to-Golgi trafficking. But *TBK1* knockout inhibited STING colocalization with Rab7 when STING in the control group began to gradually colocalize with Rab7 from 8 h to 24 h of pDNA stimulation (Figs. 4F and EV6B), which indicates that TBK1 regulates STING post-Golgi trafficking to late endosomes. *TBK1* knockout also inhibited STING protein degradation after 36 h of pDNA stimulation (Fig. EV6C). We further found that DSTYK colocalization with LAMP1 was not altered in *TBK1*-knockout cells (Fig. 4G), but *TBK1* knockout blocked STING colocalization with LAMP1 and DSTYK without affecting STING colocalization with GRASP65 (Fig. 4G). These findings confirm that TBK1 deficiency inhibits STING post-Golgi trafficking to late endosomes, and STING, therefore, could not interact with DSTYK localized at late endosomes (Fig. 4G). BX795 treatment inhibited STING colocalization with Rab7, without affecting its colocalization with GRASP65 (Figs. 4H and EV6D). Consistently, reconstitution with WT TBK1, but not the TBK1 kinase-dead mutant TBK1-S172A, restored STING colocalization with LAMP1 (Fig. EV7). Together, these data suggest that TBK1 controls STING post-Golgi

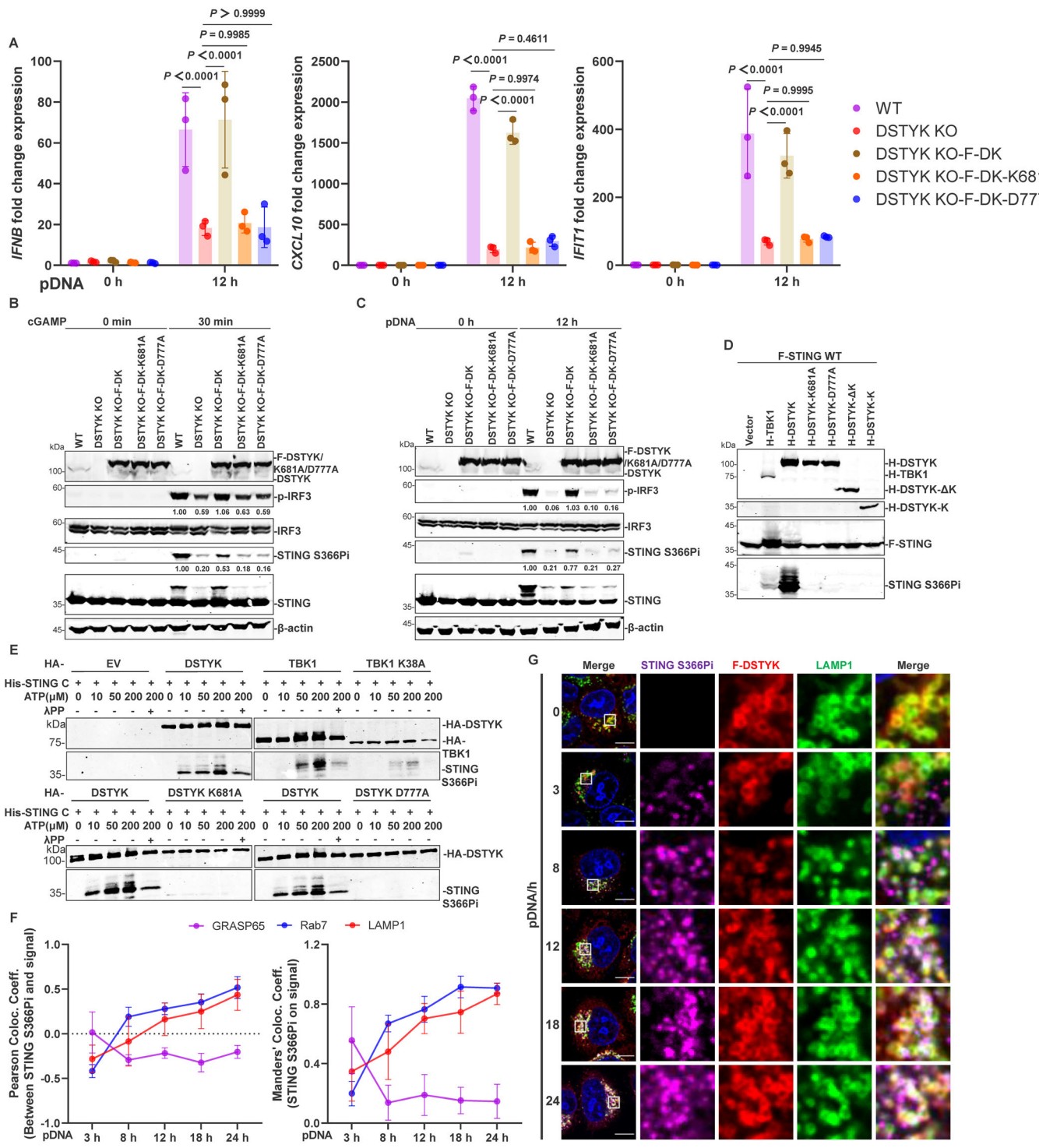

trafficking to late endosomes and STING interaction with DSTYK via its kinase activity.

It is reported that AP-1 plays a major role in TBK1-mediated STING post-Golgi trafficking (Fang et al, 2023; Liu et al, 2022) We silenced the expression of two AP-1 subunits (*AP1B1* and *AP1G1*) by shRNA (Fig. EV8A). STING colocalization with GRASP65 was not altered, but STING colocalization with LAMP1 was blocked in cells

with silenced *AP1B1* and *AP1G1* expression (Fig. EV8B), These results confirm that AP-1 deficiency blocks STING post-Golgi trafficking to late endosomes. Phosphorylation of TBK1, IRF3, and STING Ser366 induced by pDNA transfection was also suppressed in cells with silenced *AP1B1* and *AP1G1* expression (Fig. EV8C,D), confirming that blocking STING post-Golgi trafficking inhibits phosphorylation of TBK1, IRF3 and STING Ser366.

◀

**Figure 3.  DSTYK directly phosphorylates STING Ser366 at late endosomes.**

(A–C) Activation of STING signaling is dependent on DSTYK kinase activity. *DSTYK*-knockout HeLa cells stably expressing WT DSTYK (DSTYK KO-F-DK) and kinase-mutants (DSTYK KO-F-DK-K681A and DSTYK KO-F-DK-D777A) and control cells were stimulated with pDNA transfection (1 μg/mL; **A**, **C**) and cGAMP (100 nM; **B**) for the indicated amounts of time, followed by detection of *IFNB* and ISG production by quantitative PCR (**A**; $n = 3$ biological replicates) and phosphorylation of IRF3 and STING at Ser366 by western blotting (**B**, **C**). (**D**) Overexpression of DSTYK and STING promotes STING phosphorylation at Ser366. HEK293T cells were transfected with the indicated plasmids (2 μg each) for 24 h. Phosphorylated STING at Ser366 (S366Pi) and STING band shifts were detected by western blotting. (**E**) DSTYK directly phosphorylates recombinant STING Ser366 in vitro. Purified DSTYK, TBK1 and their kinase-dead mutants were incubated with recombinant STING (amino acids 153–379) under the indicated concentrations of ATP at 30 °C for 30 min, followed by detection of phosphorylated STING via western blotting. (**F**) Quantification of the colocalization coefficients between phosphorylated STING and different organelles in Figure EV5E–G. Quantification of the colocalization coefficients between phosphorylated STING and organelles was analyzed in ImageJ ($n = 6$ biological replicates). (**G**) Colocalization of phosphorylated STING, DSTYK, and LAMP1 after pDNA transfection. *DSTYK*-knockout HeLa cells stably expressing DSTYK were stimulated with pDNA (1 μg/mL) for the indicated amounts of time, followed by observation of the colocalization of phosphorylated STING, DSTYK, and LAMP1 via confocal imaging; scale bars, 10 μm. Quantification of the indicated band intensities in (**B** and **C**) was performed in ImageJ, and quantification results are labeled below the indicated bands ($n = 3$ technical replicates). Data are representative of at least three independent experiments. Data in (**A**, **F**) are shown as the mean ± SD. *P* values were determined by two-way ANOVA and the exact *P* values are shown in the figures. Source data are available online for this figure.

Combined with the above evidence, we confirm that TBK1 controls STING post-Golgi trafficking via its kinase activity, thus explaining how TBK1 regulates the interaction between STING and DSTYK and regulates the function of DSTYK in STING signaling activation at late endosomes.

## DSTYK and TBK1 both promote phosphorylation of late endosomal-localized STING

To further explore the relationship between DSTYK and TBK1 in phosphorylating STING at late endosomes, we first examined whether TBK1 can form complexes with DSTYK and STING by immunofluorescence microscopy. Colocalization among TBK1, STING, and LAMP1 after 24 h of pDNA stimulation indicates that STING can recruit TBK1 at late endosomes (Fig. 5A), and colocalization among DSTYK, TBK1, and STING further indicates that DSTYK, TBK1, and STING can form complexes at late endosomes (Fig. 5B).

Based on the above, we further fused STING C-terminus (amino acids 139-379) to late endosomal-localized TMEM192 (TMEM192-STING) to mimic sustained late endosomal-localized STING, which could eliminate the need for STING trafficking (Gentili et al, 2023). We stably expressed Flag-TMEM192-STING in HeLa cells and detected the function of TBK1 and DSTYK in the phosphorylation of TMEM192-STING Ser366. Stably-expressed TMEM192-STING showed colocalization with Rab7 both at rest and after pDNA stimulation (Fig. 5C), confirming its sustained late endosomal-localization. Stably-expressed TMEM192-STING showed spontaneous S366Pi at rest (Fig. 5D–F), which was consistent with previous report (Gentili et al, 2023), indicating that sustained late endosomal-localized STING get spontaneous activated, and pDNA stimulation strengthen S366Pi (Fig. 5D–F). *TBK1* knockout partially inhibited the phosphorylation of TMEM192-STING both at rest and after pDNA stimulation (Fig. 5D,E), confirming that TBK1 can promote phosphorylation of STING at late endosomes; *DSTYK* and *TBK1* double knockout (DKO) cells and *DSTYK* knockdown in *TBK1* knockout cells both further abolished the phosphorylation (Fig. 5D,E). The phosphorylation of TMEM192-STING in *DSTYK* knockout cells and DKO cells showed similar results (Fig. 5F). *DSTYK* knockout inhibited the phosphorylation of TMEM192-STING, and DKO further abolished the phosphorylation (Fig. 5F). These results demonstrates that DSTYK and TBK1 both promote phosphorylation of STING

Ser366 at late endosomes, and DSTYK mediated-STING S366Pi is independent of TBK1 after STING reaches late endosomes.

## DSTYK promotes host defenses against DNA virus infection in primary cells and in vivo

To gain insight into the importance of DSTYK in host defenses against viral infection in vivo, we investigated the antiviral immune response in *Dstyk*<sup>fl/fl</sup> and *Dstyk*<sup>−/−</sup> mice. Consistent with our results in human cells, peritoneal macrophages (PMs) and mouse lung fibroblasts (MLFs) from *Dstyk*<sup>−/−</sup> mice demonstrated restricted induction of the expression of *IFNB* and *IL6* after pDNA stimulation, but not after SeV infection (Fig. 6A–D). Additionally, an increase in HSV-1 titer was observed in *Dstyk*<sup>−/−</sup> MLFs (Fig. 6E). We next infected *Dstyk*<sup>fl/fl</sup> and *Dstyk*<sup>−/−</sup> mice with VACV and, as expected, observed decreased production of *IFNB* and *IL6* in tissues from *Dstyk*<sup>−/−</sup> mice infected with VACV relative to infected tissues from *Dstyk*<sup>fl/fl</sup> mice (Fig. 6F,G), and no differences in body weight were detected between groups (Fig. 6H). These results confirm that DSTYK promotes STING-mediated host defenses against infection with DNA viruses in primary cells and in vivo.

## Discussion

As the central protein in the innate immune response against cytosolic DNA, STING is heavily subject to diverse mechanisms of regulation. Post-translational modifications such as phosphorylation play important roles in regulating STING activity and ensure its proper activation. Phosphorylation of STING at Ser366 by TBK1 serves as a docking site for the recruitment of IRF3 (Liu et al, 2015; Tanaka and Chen, 2012), which is required for the production of IFNs and is an important regulator of the immune response to DNA viruses. Besides, ULK1 is reported to negatively regulate the activation of STING signaling by phosphorylating STING at Ser366 (Konno et al, 2013). It remains unknown whether there are other kinases that can also phosphorylate STING at Ser366 to promote STING signaling. Here, we identified DSTYK as a kinase that directly phosphorylates STING at Ser366 to positively regulate STING signaling. We identified the function of DSTYK in the innate immune response against pathogenic DNA, and we also identified STING as the substrate of DSTYK activity. The direct phosphorylation was observed via in vitro kinase assay (Fig. 3E),

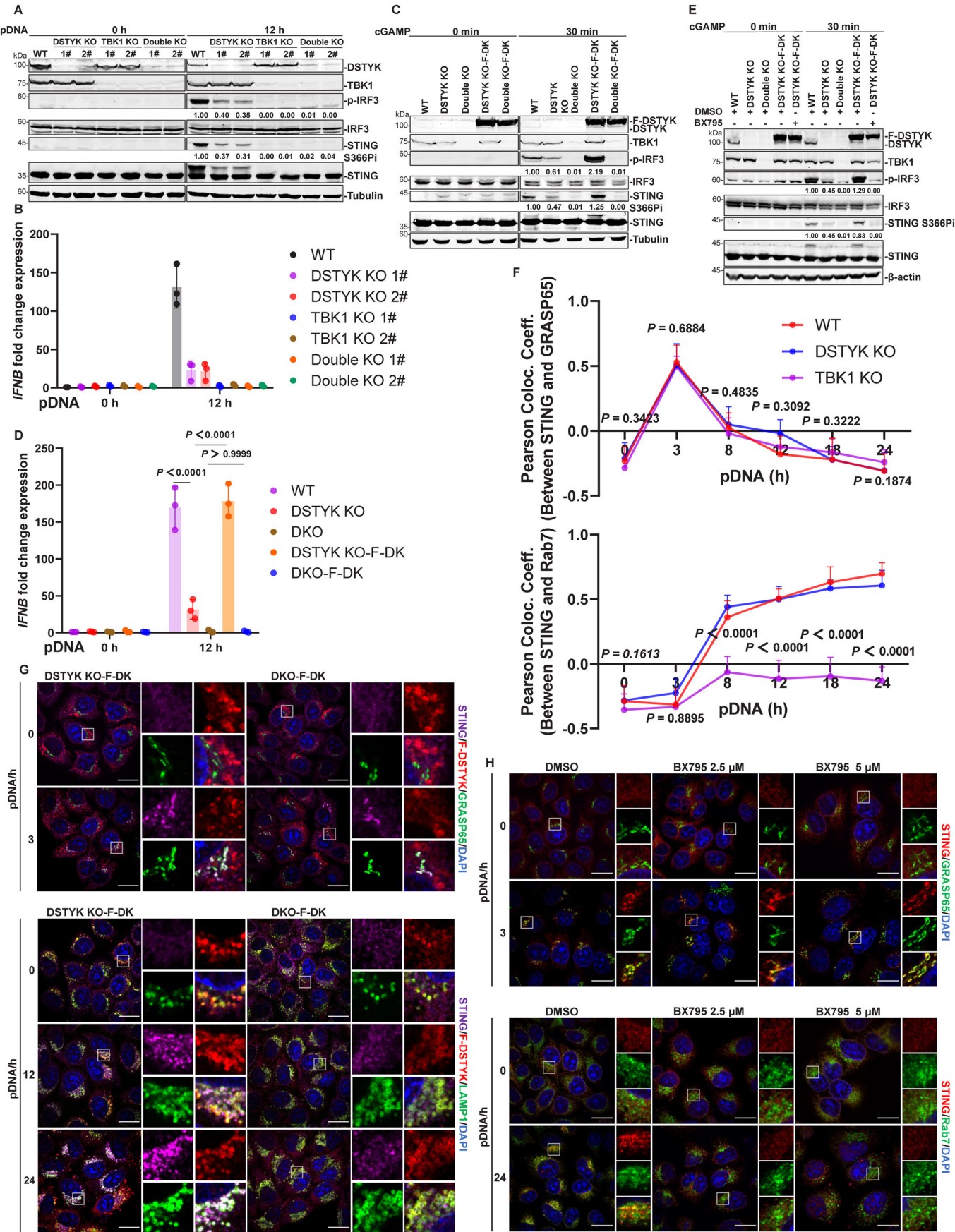

◄ **Figure 4. TBK1 regulates the function of DSTYK to STING via controlling STING post-Golgi trafficking.**

(A, B) TBK1 is essential for the activation of STING signaling. *DSTYK*-knockout HeLa cells, *TBK1*-knockout HeLa cells, DKO HeLa cells and control cells were stimulated with pDNA transfection (1 μg/mL) for 12 h, followed by detection of phosphorylation of IRF3 and STING at Ser366 via western blotting (A) and *IFNB* production via quantitative PCR (B; $n = 3$ biological replicates). (C, D) TBK1 regulates the function of DSTYK to STING. *DSTYK*-knockout HeLa cells stably expressing DSTYK (DSTYK KO-F-DK), DKO HeLa cells stably expressing DSTYK (DKO-F-DK) and control cells were stimulated with pDNA transfection (1 μg/mL) for 12 h, followed by detection of phosphorylation of IRF3 and STING at Ser366 via western blotting (C) and *IFNB* production via quantitative PCR (D; $n = 3$ biological replicates). (E) Inhibition of TBK1 kinase activity restricts the function of DSTYK. *DSTYK*-knockout HeLa cells stably expressing DSTYK (DSTYK KO-F-DK) and other control cells were pretreated with BX795 (10 μM) for 4 h and then stimulated with cGAMP (100 nM) for 30 min, followed by detection of phosphorylation of IRF3 and STING at Ser366 via western blotting. (F) Quantification of the colocalization coefficients between STING and different organelles in Fig. EV6A,B. Quantification of the colocalization coefficients between STING and organelles was analyzed in ImageJ ($n = 20$ biological replicates). (G) TBK1 controls colocalization between STING and DSTYK. *DSTYK*-knockout HeLa cells stably expressing DSTYK (DSTYK KO-F-DK) and DKO HeLa cells stably expressing DSTYK (DKO-F-DK) were stimulated with pDNA transfection (1 μg/mL) for 3, 12 and 24 h, followed by confocal imaging to observe STING and DSTYK colocalization with GRASP65 and LAMP1, respectively; scale bars, 20 μm. (H) TBK1 controls STING post-Golgi trafficking via its kinase activity. WT HeLa cells were treated with DMSO, 2.5 μM BX795 and 5 μM BX795 for 1 h, then stimulated with pDNA transfection (1 μg/mL) for 3 and 24 h, followed by confocal imaging to observe STING colocalization with GRASP65 and Rab7, respectively; scale bars, 20 μm. Quantification of the indicated band intensities in (A, C, E) was performed in ImageJ, and quantification results are labeled below the indicated bands ($n = 3$ technical replicates). Data are representative of at least three independent experiments. Data in (B, D, F) are shown as the mean ± SD. *P* values were determined by two-way ANOVA and the exact *P* values are shown in the figures. Source data are available online for this figure.

indicating a direct interaction between DSTYK and STING, but this needs to be further defined by determining the cryo-EM structure of STING with DSTYK.

We confirmed that DSTYK is located at late endosomal membrane (Fig. 2E–H), where the known key signaling molecules of the innate immune pathway in response to RNA viruses (i.e., RIG-I and VISA) were not distributed. This finding could explain why DSTYK does not regulate SeV-induced innate immune pathway activation (Figs. 1E and EV2D, 3C,F).

Activation of STING signaling requires trafficking from the ER to the ER–Golgi intermediate compartment and the Golgi, which is reported to be essential for STING puncta formation, a process fundamental to the activation of type I IFN responses via TBK1 at the *trans*-Golgi network (TGN) (Kemmoku et al, 2024; Mukai et al, 2016). STING post-Golgi trafficking were once reported to be responsible for its protein degradation and thus negatively regulate STING signaling (Gonugunta et al, 2017). Recently, PI4P-driven, ARF1-dependent STING TGN–endosomal trafficking was found to be important for the activation of STING signaling (Fang et al, 2023), and Gui et al, proposed a model for activation of STING signaling at late endosomes (Gui et al, 2019). Additionally, sustained late endosomal-localization of TMEM192-STING leads to spontaneous activation of STING signaling in the absence of ligand (Gentili et al, 2023). These has suggested that late endosome can act as an important platform for the activation of STING signaling. In this research, we found that DSTYK positively regulates the activation of STING signaling at late endosomes (Figs. 1 and 2). *DSTYK* knockout cells showed similar STING trafficking kinetics and STING protein level with control group after pDNA stimulation (Fig. 4F and EV6A–C), confirming that DSTYK does not affect STING trafficking from ER to late endosomes, and DSTYK does not affect STING degradation either. Besides, quantifying the colocalization of phosphorylated STING with different organelles showed that phosphorylated STING colocalized with late endosome markers (Rab7 and LAMP1) from 8 h of stimulation onward (Fig. 3F), suggesting that STING can get phosphorylated at late endosomes, which further indicates the importance of late endosomes in the activation of STING signaling.

In this research, we demonstrate that TBK1 regulates the function of DSTYK in the activation of STING signaling via its kinase activity. Specifically, we report here a role for the kinase activity of TBK1 in orchestrating STING post-Golgi trafficking to late endosomes (Figs. 4F,H and EV6, 7). Lack of TBK1 kinase activity abolished STING post-Golgi trafficking, thus inhibiting the interaction between STING and DSTYK (Figs. 4G,H and EV7), in which case DSTYK cannot phosphorylate STING and regulate the activation of STING signaling. It is reported that STING phosphorylation by TBK1 promotes the recruitment of AP-1 to control STING post-Golgi trafficking (Liu et al, 2022), but previous reports are contradictory regarding whether AP-1 acts as a positive or negative regulator in STING signaling (Fang et al, 2023; Liu et al, 2022). In this research, we found that blocking the post-Golgi trafficking of STING by silencing the expression of AP-1 significantly inhibited pDNA transfection-induced phosphorylation of TBK1, IRF3, and STING Ser366 (Fig. EV8A–D), which is consistent with the previous report (Fang et al, 2023).

Our results further showed that STING colocalized with both DSTYK and TBK1 at late endosomes (Fig. 5A,B), indicating that DSTYK, TBK, and STING can form complexes at late endosomes, and functional detection using sustained late endosomal-localized TMEM192-STING chimeric protein further demonstrate that DSTYK and TBK1 can both phosphorylate STING Ser366 at late endosomes (Fig. 5D–F). Stably-expressed TMEM192-STING showed spontaneous S366Pi at rest, which was consistent with previous report (Gentili et al, 2023), and cGAMP binding may change the conformation of TMEM192-STING, thus enhancing the phosphorylation of TMEM192-STING. However, our co-IP results showed that DSTYK did not interact with TBK1 in STING-non-expressing HEK293T cells (Fig. 2C). This indicates that DSTYK and TBK1 may not have a direct interaction in the absence of STING, but they can form complexes with STING at late endosomes.

Besides, we confirmed that DSTYK deficiency inhibited cytosolic DNA- and cGAMP-induced TBK1 phosphorylation (Figs. 1C,D, 2A,B and EV2E, 3A,B,D,E), indicating that DSTYK can also promote the activation of TBK1. This finding could be explained by the fact that TBK1 is more inclined to bind phosphorylated STING (Kemmoku et al, 2024; Zhao et al, 2019). STING gets phosphorylated by DSTYK after reaching late endosomes, and the increased level of STING phosphorylation allows for the recruitment of more TBK1, thereby enhancing TBK1 activation. DSTYK deficiency leads to a reduction in the phosphorylation of STING Ser366, which may

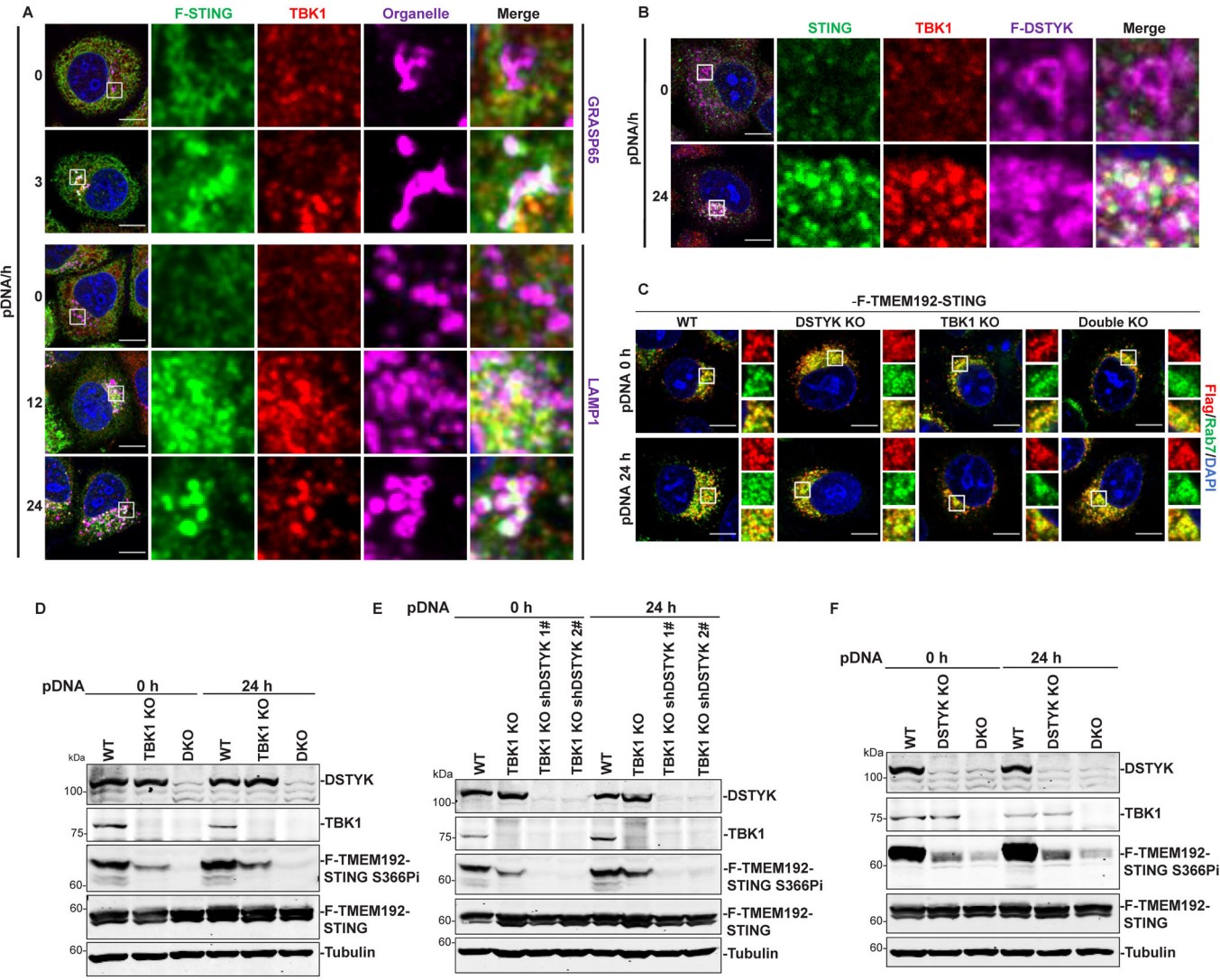

**Figure 5. DSTYK and TBK1 both promote phosphorylation of late endosomal-localized STING.**

(A) TBK1 colocalizes with STING from Golgi to late endosomes. *STING* knockout HeLa cells stably-expressing Flag-STING were stimulated with pDNA transfection (1 μg/mL) for the indicated amounts of time, and followed by observing STING, TBK1 colocalization with GRASP65 and LAMP1. Scale bars, 10 μm. (B) TBK1 colocalizes with STING and DSTYK at late endosomes. *DSTYK* knockout HeLa cells stably-expressing Flag-DSTYK were stimulated with pDNA transfection (1 μg/mL) for 24 h, and followed by observing STING, TBK1 colocalization with DSTYK. Scale bars, 10 μm. (C) Stably-expressed TMEM192-STING localizes at late endosomes both at rest and after pDNA stimulation. *TBK1*-knockout HeLa cells, *DSTYK*-knockout HeLa cells, DKO HeLa cells and control cells stably-expressing Flag-TMEM192-STING were stimulated with pDNA (1 μg/mL) for 24 h, and followed by observing Flag-TMEM192-STING colocalization with Rab7. Scale bars, 10 μm. (D, E) TBK1 promotes phosphorylation of STING at late endosomes. *TBK1*-knockout HeLa cells, DKO HeLa cells (D) (or *DSTYK* knockdown in *TBK1*-knockout HeLa cells, E) and control cells stably-expressing Flag-TMEM192-STING were stimulated with pDNA (1 μg/mL) for 24 h, followed by detection of phosphorylation of STING at Ser366 via western blotting. (F) DSTYK promotes phosphorylation of STING at late endosomes. *DSTYK*-knockout HeLa cells, DKO HeLa cells and control cells stably-expressing Flag-TMEM192-STING were stimulated with pDNA (1 μg/mL) for 24 h, followed by detection of phosphorylation of STING at Ser366 via western blotting. Data are representative of at least three independent experiments. Source data are available online for this figure.

weaken the recruitment of TBK1 to STING and thereby diminishing TBK1 activation. Consequently, TBK1's function in phosphorylating and activating IRF3 is also inhibited.

Based on all the above, we suggest the following model of STING signaling. After binding cGAMP, STING moves to the Golgi where it recruits TBK1, and TBK1 recruitment promotes the post-Golgi trafficking of STING via its kinase activity. TBK1 and late endosomal-localized DSTYK can both directly phosphorylate

late endosomal-localized STING at Ser366, thus further promoting maximized activation of STING signaling (Fig. EV8E).

In summary, we identified DSTYK as a kinase that directly phosphorylates STING at Ser366 at late endosomes to activate STING signaling. Further dissection of the mechanisms underlying the activation of STING signaling at late endosomes may lead to the design of specific therapeutic strategies toward modulating STING signaling in the context of STING-related immune diseases.

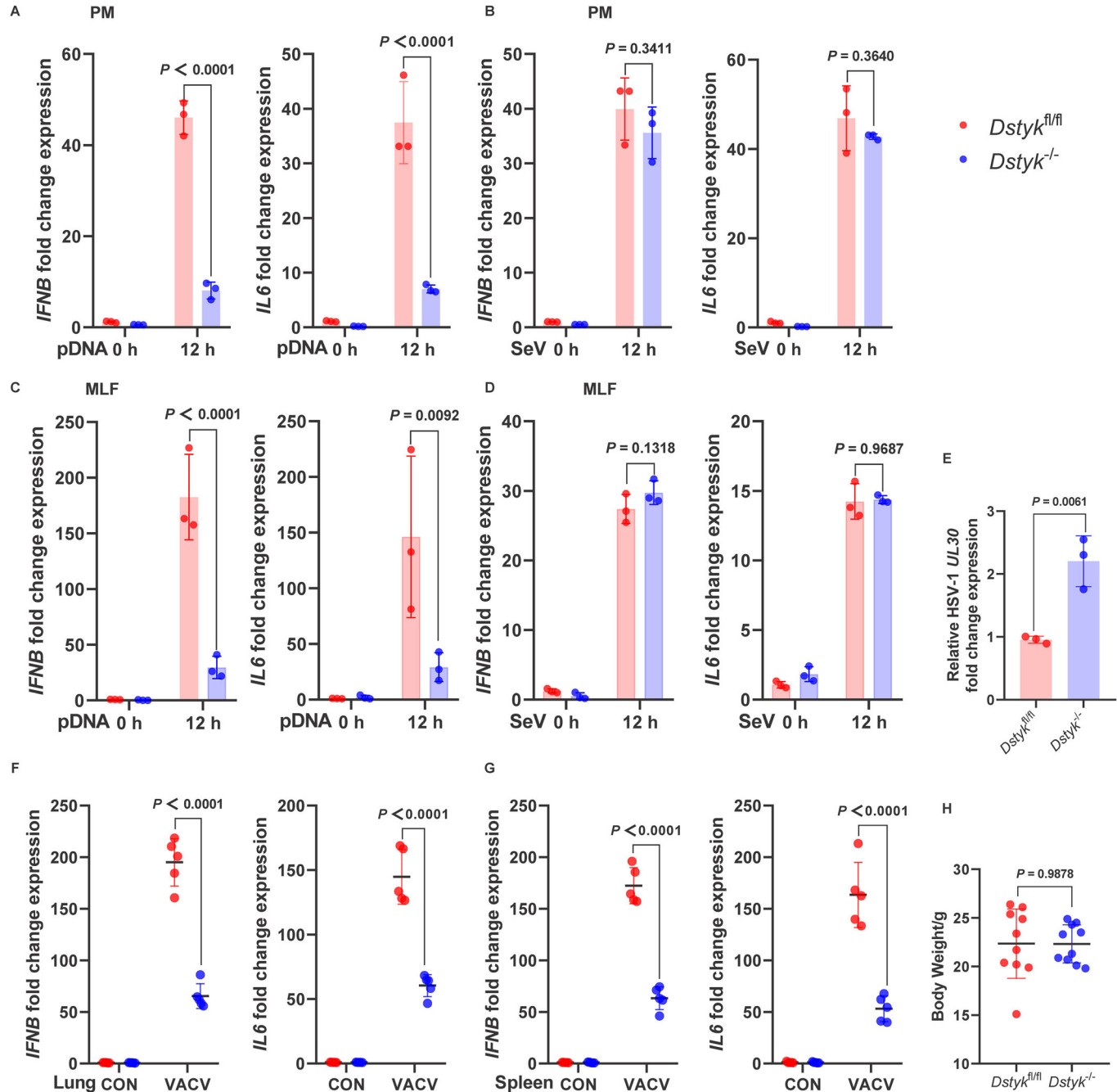

**Figure 6. DSTYK promotes host defenses against DNA virus infection in primary cells and in vivo.**

(A–D) *Dstyk* knockout in primary mouse cells inhibits cytosolic DNA-triggered *IFNB* and *IL6* production. *Dstyk*-knockout PMs (A, B) and MLFs (C, D) were stimulated with pDNA (1 μg/mL; A, C) and SeV (B, D) for 12 h, followed by detection of *IFNB* and *IL6* production via quantitative PCR (*n* = 3 biological replicates). (E) *Dstyk* knockout in MLFs promotes HSV-1 replication. *Dstyk*-knockout MLFs and control cells were stimulated with HSV-1 for 24 h, and HSV-1 *UL30* production was detected by quantitative PCR (*n* = 3 biological replicates). (F, G) *Dstyk* knockout inhibits VACV-induced *IFNB* and *IL6* production in vivo. *Dstyk*^fl/fl^ mice and *Dstyk*^−/−^ mice were infected with a lethal dose of VACV intranasally, followed by detection of the production of *IFNB* and *IL6* in the indicated tissues 4 days after infection (*n* = 5 biological replicates). (H) No body weight differences between WT and *Dstyk*-knockout mice (F, G). Data in (A–H) are shown as mean ± SD. Data are representative of at least two independent experiments. *P* values in (E, H) were determined by Student's *t*-tests, and *P* values in (A–D, F, G) were determined by two-way ANOVA and the exact *P* values are shown in the figures. Source data are available online for this figure.

# Methods

### Reagents and tools table

| Reagent/Resource | Reference or Source | Identifier or Catalog Number |
| --- | --- | --- |
| **Experimental models** | | |
| C57BL/6J (*M. musculus*) | Cyagen | KOCMP-213452-Dstyk-B6J-VA |
| **Recombinant DNA** | | |
| pSIN-Flag-DSTYK | This manuscript | N/A |
| pSIN-Flag-DSTYK-K681A | This manuscript | N/A |
| pSIN-Flag-DSTYK-D777A | This manuscript | N/A |
| pSIN-Flag-TBK1 | This manuscript | N/A |
| pSIN-Flag-TBK1-S172A | This manuscript | N/A |
| pSIN-Flag-STING | This manuscript | N/A |
| pSIN-Flag-TMEM192-STING | This manuscript | N/A |
| pRK-HA-DSTYK | This manuscript | N/A |
| pRK-HA-DSTYK-K681A | This manuscript | N/A |
| pRK-HA-DSTYK-D777A | This manuscript | N/A |
| pRK-HA-DSTYK-ΔK | This manuscript | N/A |
| pRK-HA-DSTYK-K | This manuscript | N/A |
| pRK-HA-cGAS | This manuscript | N/A |
| pRK-HA-RIG-I | This manuscript | N/A |
| pRK-HA-VISA | Provided by H. Shu (Wuhan University) | N/A |
| pRK-HA-TBK1 | Provided by H. Shu (Wuhan University) | N/A |
| pRK-HA-TBK1-K38A | This manuscript | N/A |
| pCMV14-Flag-STING | This manuscript | N/A |
| **Antibodies** | | |
| Mouse monoclonal anti-HA | Sigma-Aldrich | H9658 |
| Mouse monoclonal anti-Flag | Sigma Aldrich | F1804 |
| Mouse monoclonal anti-β-actin | Proteintech | 66009-1-Ig |
| Mouse monoclonal anti-GAPDH | Proteintech | 60004-1-Ig |
| Mouse monoclonal anti-α-tubulin | Proteintech | 66031-1-Ig |
| Mouse monoclonal anti-calnexin | Proteintech | 66903-1 |
| Mouse monoclonal anti-STING | Proteintech | 66680-1-Ig |
| Mouse monoclonal anti-GRASP65 | Santa Cruz Biotechnology | sc-374423 |
| Mouse monoclonal anti-LAMP1 | Santa Cruz Biotechnology | sc-20011 |
| Mouse monoclonal anti-Rab7 | Santa Cruz Biotechnology | sc-376362 |
| Mouse monoclonal anti-EEA1 | Santa Cruz Biotechnology | sc-137130 |
| Rabbit monoclonal antibodies to phospho-IRF3 (S386) | Abcam | ab76493 |
| Rabbit monoclonal anti-phospho-TBK1/NAK (S172) | Abcam | ab109272 |
| Rabbit monoclonal anti-phospho-NF-κB p65 (Ser536) | Cell Signaling Technology | 3033 |
| Rabbit monoclonal anti-phospho-IκBα (Ser32) | Cell Signaling Technology | 2859 |
| Rabbit monoclonal anti-phospho-STING (Ser366) | Cell Signaling Technology | 19781 |
| Rabbit monoclonal anti-TBK1/NAK | Abcam | ab40676 |
| Rabbit monoclonal anti-NF-κB p65 | Cell Signaling Technology | 4764 |
| Rabbit polyclonal anti-STING | ABclonal | A3575 |
| Rabbit polyclonal anti-STING | Proteintech | 19851-1-AP |
| Rabbit polyclonal anti-Flag | Proteintech | 20543-1-AP |
| Rabbit polyclonal anti-DSTYK | Proteintech | 20102-1-AP |
| Rabbit polyclonal anti-IRF3 | Proteintech | 11312-1-AP |
| Roat polyclonal anti-Flag | Abcam | ab95045 |
| Alexa Fluor 488-labeled goat anti-mouse IgG (H + L) cross-adsorbed secondary | Thermo Fisher Scientific | A11001 |
| Alexa Fluor 594-labeled donkey anti-rabbit IgG (H + L) highly cross-adsorbed secondary | Thermo Fisher Scientific | A21207 |
| Alexa Fluor 647-labeled donkey anti-goat IgG (H + L) cross-adsorbed secondary | Thermo Fisher Scientific | A21447 |
| **Oligonucleotides and other sequence-based reagents** | | |
| gRNA for human DSTYK and TBK1 | This manuscript, shown below | |
| shRNA for human DSTYK, AP1B1 and AP1G1 | This manuscript, shown below | |
| Primers for qPCR | This manuscript, shown below | |
| **Chemicals, Enzymes and other reagents** | | |
| 2'3'-cGAMP | InvivoGen | tlrl-nacga23m |
| LPS | Sigma | L7895 |
| poly (I:C) | Amersham | 27-4732-01 |
| Pam2CSK4 | Invivogen | tlrl-pm2s-1 |
| Phorbol myristate acetate | Invivogen | tlrl-pma |
| Digitonin | Sigma | 300410 |
| Opti-MEM | Gibco | 51985034 |
| BX795 | TargetMol | T1830 |
| LipoMax | SUNGEN | 32012 |
| liposomal transfection reagent | YEASEN | 40802ES03 |
| Polyethylenimine | polysciences | 24765 |
| Puromycin | VWR Life Science | 97064-280 |
| Quantitative PCR SYBR Green master mix | YEASEN | 11201ES08 |
| TriQuick reagent | Solarbio | R1100 |
| Fast Mutagenesis System | Transgen | FM111-01 |
| Thioglycollate medium | BD | 211716 |

| Reagent/Resource | Reference or Source | Identifier or Catalog Number |
|---|---|---|
| **Software** | | |
| Image J | https://imagej.net/ij/ | |
| GraphPad Prism | https://www.graphpad.com/ | |

## Mice

Male and female C57BL/6 mice were sex matched and studied at 6–10 weeks of age. All mice were bred and housed under specific pathogen-free conditions. All experiments were conducted in accordance with the National Institute of Health Guide for Care and Use of Laboratory Animals. Animal protocols were approved by the Institutional Animal Care and Use Committee at Peking University (LSC-ChenDY-1).

$Dstyk^{-/-}$ mice of a C57BL/6 background were generated by Cyagen Biosciences.

## Cell lines and cultures

Primary mouse PMs were collected 5–7 d after intraperitoneal injection of 3% thioglycollate medium into 6- to 8-week-old C57BL/6J mice and were cultured in DMEM supplemented with 5% fetal bovine serum (FBS), 5 µg/mL penicillin and 10 µg/mL streptomycin for use.

Lung tissue from 6- to 8-week-old C57BL/6J mice was cut into small pieces and digested with 0.1% collagenase type I in D-Hanks buffer at 37 °C for 3 h. Digested fibroblasts were centrifuged and cultured in DMEM containing 10% FBS, 5 µg/mL penicillin, and 10 µg/mL streptomycin for 3 d. Nonadherent cells were washed off, and adherent cells were cultured in DMEM containing 10% FBS, 5 µg/mL penicillin, and 10 µg/mL streptomycin and used as MLFs.

HEK293T, HeLa, Vero (H. Shu, Wuhan University) and HT1080 (Cell Resource Center, Institute of Basic Medicine, Chinese Academy of Medical Sciences/Peking Union Medical College) cells were cultured in DMEM supplemented with 10% FBS, 5 µg/mL penicillin and 10 µg/mL streptomycin at 37 °C and 5% $CO_2$. THP1 (Cell Resource Center, Institute of Basic Medicine, Chinese Academy of Medical Sciences/Peking Union Medical College) were cultured in RPMI-1640 medium supplemented with 10% FBS, 5 µg/mL penicillin and 10 µg/mL streptomycin at 37 °C and 5% $CO_2$. For the differentiation of monocytic THP1 into macrophages, THP1 was treated with phorbol myristate acetate (100 ng/mL) for 12 h, then wash the cells with D-Hanks and incubate the cells with fresh 10% FBS RPMI-1640 medium for 24 h. Fully differentiated macrophages can be used for experiments.

## Transfection

PMs, MLFs, HeLa cells, and HT1080 cells were transfected using LipoMax or Liposomal transfection reagent, and HEK293T cells were transfected using polyetherimide. Transfections were performed following the manufacturer's instructions.

## Virus amplification

SeV, HSV-1 WT F strain (H. Shu, Wuhan University) and VACV Western Reserve strain (M. Fang, Chinese Academy of Medical Sciences) were obtained from the indicated sources. SeV was amplified using specific pathogen-free chicken eggs, and HSV-1 and VACV were amplified on Vero cells. Cells were collected after 48 h of infection and resuspended in phosphate buffer, after which, freezing and liquification were repeated three times to release the amplified virus.

## Analysis of the expression profile of DSTYK in immune cells and human cell lines

Values for the expression of human and mouse DSTYK mRNA in various immune cells were extracted from Gene Skyline from the Immunological Genome Project. (ImmGen; http://rstats.immgen.org/Skyline/skyline.html).

Values for the expression of human DSTYK mRNA in human cell lines were extracted from The Human Protein Atlas. (https://www.proteinatlas.org/ENSG00000133059-DSTYK/cell+line).

## Generation of stable knockdown or overexpression cells

shRNA lentivirus-expressing plasmids were transfected into HEK293T cells with three packaging plasmids to generate stable knockdown cells. pSIN-puro plasmids with different inserted target genes were transfected into HEK293T cells together with packing plasmids. After 12 h, cells were incubated with DMEM supplemented with 15% FBS for 36 h. Medium containing the recombinant lentivirus was centrifuged to remove cell debris, and the supernatant was then used to generate stable knockdown and overexpressing cell lines. Cells were exposed to the indicated lentivirus supernatant in the presence of polybrene (8 mg/mL) for 48 h, followed by selection with puromycin (1 mg/mL) for further use.

The following shRNA sequences were used:
human *DSTYK* 1: 5′-GCACTGGAATGATCTGGCTTT-3′;
human *DSTYK* 2: 5′-GCTTTGGAATTTCACTATATG-3′;
human *AP1B1* 1: 5′-CCATTGTGAAACTCTTTCTAA-3′;
human *AP1B1* 2: 5′-CGTTGACAAGATCACAGAGTA-3′;
human *AP1G1* 1: 5′-GCCCTGGTAAATGGGAATAAT-3′;
human *AP1G1* 2: 5′-GCAGGAAGTTATGTTCGTGAT-3′.

## CRISPR–Cas9-mediated genome editing

Construct guide RNA targeted *DSTYK* and *TBK1*, respectively, into a lenti-CRISPR vector. Lenti-CRISPR *DSTYK* single guide RNA plasmid and lenti-CRISPR-TBK1 single guide RNA plasmid were transfected into HeLa cells by using Lipomax. Cells were incubated with DMEM supplemented with 15% FBS for another 12 h, followed by selection with 1 mg/mL puromycin. Flow cytometry was used to separate individual cells. Efficient knockout was verified by western blotting. The control group consisted of monoclonal cells with no genes knocked out and had activity similar to that of WT cells.

The following guide RNA sequences were used: human *DSTYK*: 5′-GAATGATCCGCGAGCTGTGC-3′; human *TBK1*: 5′-GATGAAGATCAACCTGGAAG-3′.

## Stimulation with 2′3′-cGAMP

HeLa cells were incubated in reactions containing Opti-MEM, 10 mg/mL digitonin and 100 nM 2′3′-cGAMP at 37 °C for the indicated lengths of time before further use.

## Treatment with λ-phosphatase

Protein samples were added with $MgCl_2$ buffer, followed by treatment with λ-phosphatase (1 μL of λ-phosphatase treatment per 50 μL of sample) at 37 °C for 1 h.

## Coimmunoprecipitation and western blotting

HEK293T and HeLa cells were subjected to the indicated stimulation or transfected with the indicated plasmids before being lysed in Triton X-100 lysis buffer (20 mM Tris-HCl (pH 7.4), 150 mM NaCl, 1 mM EDTA, 1% Triton X-100, 10 μg/mL aprotinin, 10 μg/mL leupeptin and 1 mM phenylmethylsulphonyl fluoride). Immunoprecipitation samples were incubated with Protein G Sepharose beads (GE Healthcare) and the indicated antibodies in lysate at 4 °C. Sepharose beads were then washed with lysis buffer three times, and the precipitates were boiled for 10 min at 95 °C and analyzed by SDS–PAGE, followed by standard immunoblotting. Final results were visualized using an Odyssey infrared imaging system (LICOR). The relevant immunoblot band intensities were quantified using ImageJ.

## Protein purification and in vitro kinase assay

Human STING (amino acids 153–379) was purified from *Escherichia coli*, His tagged, transformed in *E. coli* and induced with isopropyl β-D-1-thiogalactopyranoside at 37 °C for 4 h. Bacteria were then collected by centrifugation and lysed by sonication in IB wash buffer (15 mM Tris-HCl (pH 7.5), 10 mM EDTA and 1% TritonX-100). Inclusion bodies were collected by centrifugation after the addition of phenyl-methylsulfonyl fluoride. Inclusion bodies were then resuspended and lysed in IB solubilization buffer (5 mM CAPS (pH 11)), 3% sodium lauroyl sarcosine (SLS) and 1 M dithio-threitol (DTT), followed by centrifugation and dialysis of the clear lysate.

Human DSTYK and human TBK1 protein were purified from HEK293T cells. Full-length human DSTYK, TBK1 and their kinase-dead mutants with fused Flag tag or HA tag were expressed in HEK293T cells for 24 h. Proteins were extracted in Triton X-100 lysis buffer, and proteins in the supernatant were captured by Protein G Sepharose beads and the indicated antibodies equilibrated with lysis buffer, followed by washing with lysis buffer and kinase buffer (25 mM Tris-HCl (pH 7.5), 5 mM β-glycerophosphate, 2 mM DTT, 0.1 mM $Na_3VO_4$ and 10 mM $MgCl_2$).

For the in vitro kinase assays, proteins bound to the beads were resuspended with kinase buffer, and different concentrations of STING protein and ATP were added according to the experimental requirements. The reaction was performed at 30 °C for 30 min, 2× SDS sample buffer was added, and the reaction was terminated by heating at 95 °C for 10 min. The precipitates were analyzed by SDS–PAGE.

## Immunofluorescence microscopy

Cells grown on glass coverslips were stimulated as indicated, washed with PBS and fixed in absolute methanol for 20 min at −20 °C, followed by washing with PBS. Cells on the slides were incubated with 5% bovine serum albumin in PBS for 30 min and incubated with primary and secondary antibodies. Nuclei were stained with 4,6-diamidino-2-phenylindole at room temperature for 10 min. Confocal images were acquired on a Nikon Live SR CSU W1 confocal microscope using a ×100/1.45-NA objective. Super-resolution images were acquired on a CSR high-sensitivity structured illumination microscope, and colocalization of relevant proteins was analyzed using ImageJ.

## Quantitative real-time PCR

Total RNA from cell lines and mouse tissues was isolated with TriQuick reagent (Solarbio) according to the manufacturer's instructions. Total RNA was converted into cDNA with oligo(T) primers and reverse transcriptase (Fermentas). Quantitative real-time PCR was completed using SYBR Green PCR master mix on a Light-Cycle 96 System (Roche), and gene expression levels were calculated by calculating the accumulation index ($2^{-\Delta\Delta Ct}$).

The following primers for were used:
human *IFNB* (forward): 5′-CCAACAAGTGTCTCCTCCAA-3′
human *IFNB* (reverse): 5′-ATAGTCTCATTCCAGCCAGT-3′
human *CXCL10* (forward): 5′-GTGGCATTCAAGGAGTACCTC-3′
human *CXCL10* (reverse): 5′-TGATGGCCTTCGATTCTGGATT-3′
human *IFIT1* (forward): 5′-GACTGTGAGGAAGGATGGGC-3′
human *IFIT1* (reverse): 5′-TAGGCTGCCCTTTTGTAGCC-3′
human *IL6* (forward): 5′-AGAGGCACTGGCAGAAAACAAC-3′
human *IL6* (reverse): 5′-AGGCAAGTCTCCTCATTGAATCC-3′
human *ACTB* (forward): 5′-ACGTGGACATCCGCAAAGAC-3′
human *ACTB* (reverse): 5′-CAAGAAAGGGTGTAACG-CAACTA-3′
human *AP1B1* (forward): 5′-TCTTCCGCAAGTACCCCAAC-3′
human *AP1B1* (reverse): 5′-CAGCTCCTGGGTCTCTGTTG-3′
human *AP1G1* (forward): 5′-TGCTGCAATCCGGTCATCTT-3′
human *AP1G1* (reverse): 5′-GCCGAGGGTACAAAGTGCTA-3′
mouse *Ifnb* (forward): 5′-CACAGCCCTCTCCATCAACT-3′
mouse *Ifnb* (reverse): 5′-TCCCACGTCAATCTTTCCTC-3′
mouse *Il6* (forward): 5′-TCTGCAAGAGACTTCCATCCAGTTGC-3′
mouse *Il6* (reverse): 5′-AGCCTCCGACTTGTGAAGTGGT-3′
mouse *Gapdh* (forward): 5′-TGATGGGTGTGAACCACGAG-3′
mouse *Gapdh* (reverse): 5′-TAGGGCCTCTCTTGCTCAGT-3′
HSV-1 *UL30* (forward): 5′-CATCACCGACCCGGAGAGGGAC-3′
HSV-1 *UL30* (reverse): 5′-GGGCCAGGCGCTTGTTGGTGTA-3′

## Statistical analysis

Statistical analyses were performed using GraphPad Prism, and images were analyzed in ImageJ. Student's *t*-tests were performed when comparing the statistically significant differences between two groups, and experiments involving multiple tests and variables

were analyzed by analysis of variance. All statistical results are shown as the mean ± standard deviation (SD; $n \geq 3$); ns, not significant ($P > 0.05$); $*P < 0.05$; $**P < 0.01$; $***P < 0.001$; $****P < 0.0001$. No sample size estimation was predetermined.

## Data availability

This study includes no data deposited in external repositories.

The source data of this paper are collected in the following database record: biostudies:S-SCDT-10_1038-S44319-025-00394-9.

## Peer review information

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

## Acknowledgements

We thank Prof. HongBing Shu from Wuhan University, China, Prof. ZhengFan Jiang, Prof. Jianguo Chen and Prof. Junlin Teng from Peking University, China, and Prof. Min Fang from the Chinese Academy of Sciences, China for plasmids, viruses, cell lines and instruments. We also thank members from the National Center for Protein Sciences at Peking University for assistance with optical microscopy, flow cytometry and quantitative real-time PCR, in particular Dr. Siying Qin, Dr. Jun Ren and Dr. Wenling Gao for technical help with optical microscopy, Ms. Liying Du for technical help with flow cytometry and Ms. Guilan Li for technical help with quantitative real-time PCR. We also thank Dr. Guopeng Wang for suggestion. This work benefitted from data assembled by the ImmGen consortium and The Human Protein Altas. This work was supported by the National Natural Science Foundation of China (31870904).

## Author contributions

**Hao Dong**: Conceptualization; Data curation; Formal analysis; Validation; Investigation; Visualization; Methodology; Writing—original draft; Writing—review and editing. **Heng Zhang**: Conceptualization; Validation; Investigation; Visualization; Methodology. **Pu Song**: Investigation. **Yuan Hu**: Investigation. **Danying Chen**: Conceptualization; Supervision; Funding acquisition; Visualization; Methodology; Project administration; Writing—review and editing.

Source data underlying figure panels in this paper may have individual authorship assigned. Where available, figure panel/source data authorship is listed in the following database record: biostudies:S-SCDT-10_1038-S44319-025-00394-9.

## Disclosure and competing interests statement

The authors declare no competing interests.

# Expanded View Figures

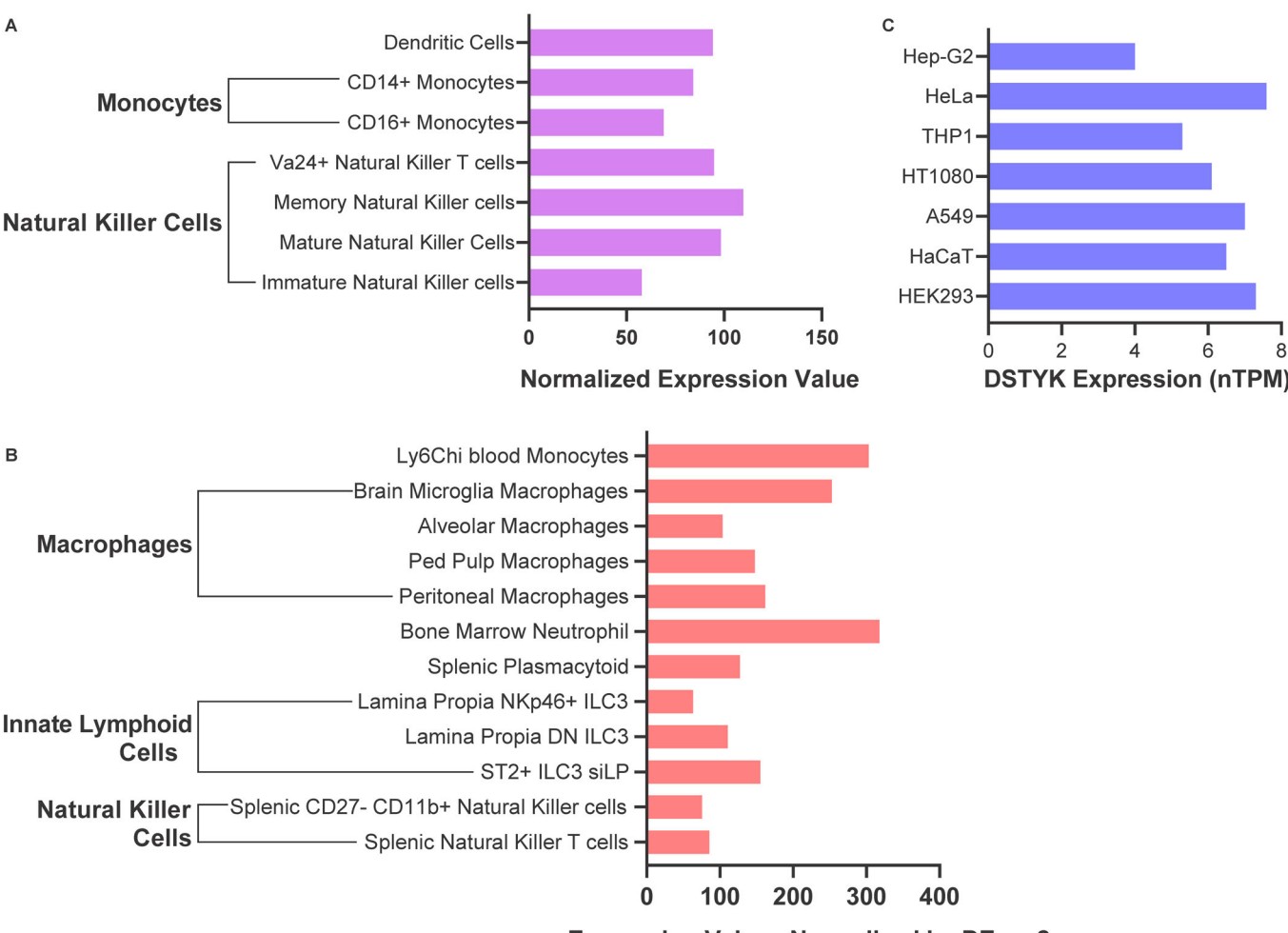

**Figure EV1.  DSTYK is widely expressed among primary immune cells and human cell lines.**

Analysis of the expression profile of human DSTYK (**A**) and mouse Dstyk (**B**) in primary innate immune cells are derived from Gene Skyline in ImmGen, including natural killer cells, innate lymphoid cells, neutrophils, dendritic cells, and macrophages. Analysis of the expression profile of human DSTYK in common human cell lines is derived from The Human Protein Atlas (**C**), including HEK293, HaCaT, A549, HT1080, THP1, HeLa, and Hep-G2.

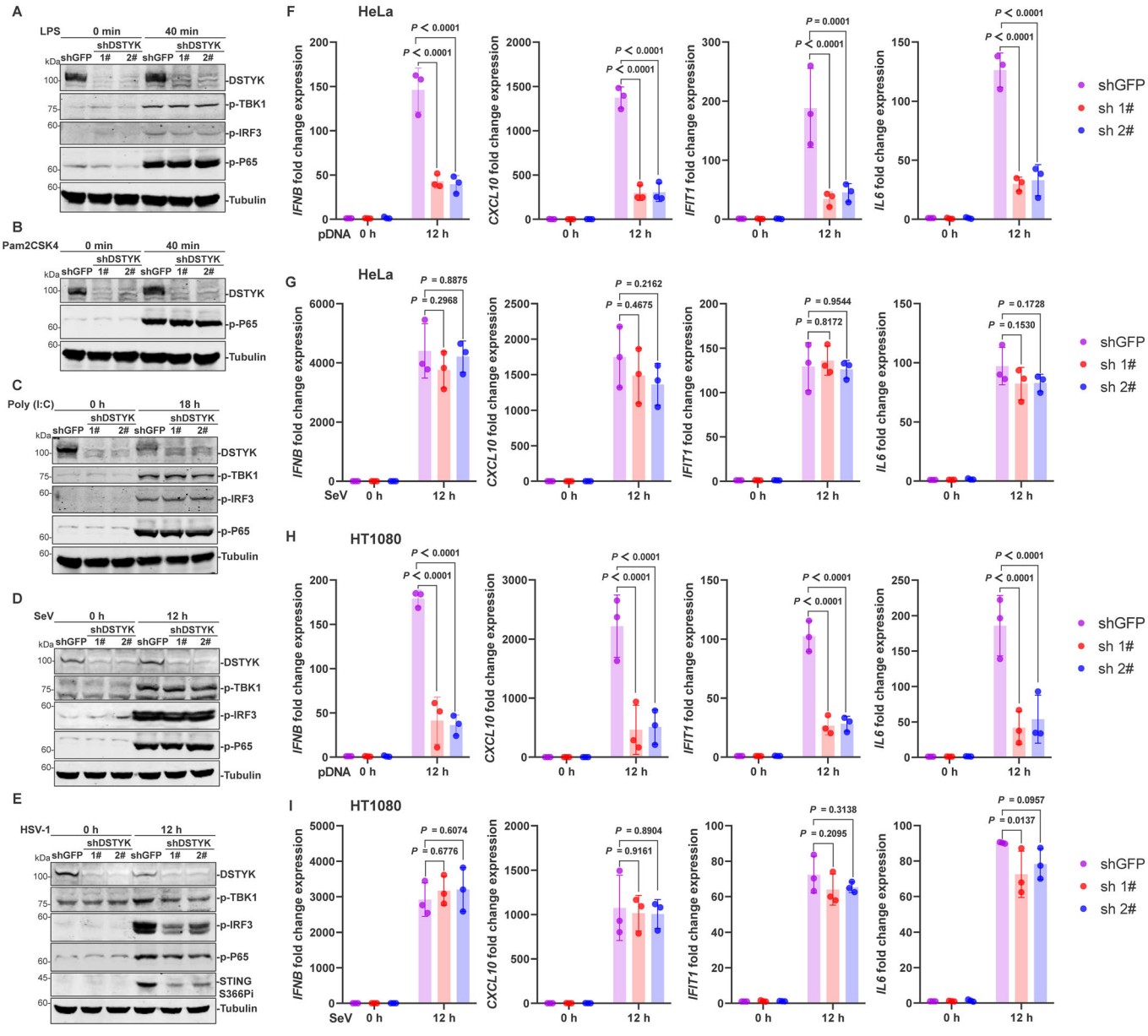

**Figure EV2.  DSTYK deficiency inhibits the cytosolic DNA-triggered cGAS–STING pathway.**

(A–E) *DSTYK*-knockdown THP1-derived macrophages and control cells (shGFP) were stimulated with LPS (1 μg/mL, **A**), Pam2CSK4 (1 ng/mL, **B**), poly (I:C) (10 μg/mL, **C**), SeV (**D**) and HSV-1 (**E**) for the indicated amounts of time, and phosphorylation of TBK1, IRF3, P65 and STING at Ser366 was detected by western blotting. (F–I) *DSTYK* knockdown in HeLa and HT1080 cells inhibits cytosolic DNA-triggered production of *IFNB*, ISGs and genes encoding proinflammatory cytokines. *DSTYK*-knockdown HeLa (**F**, **G**) and HT1080 (**H**, **I**) cells and control cells (shGFP) were stimulated with pDNA transfection (1 μg/mL; **F**, **H**) and SeV (**G**, **I**) for 12 h, and *IFNB*, ISG (*CXCL10* and *IFIT1*) and proinflammatory cytokine (*IL6*) production was detected by quantitative PCR (*n* = 3 biological replicates). Data are representative of at least three independent experiments. Data in (**F–I**) are shown as the mean ± SD. *P* values were determined by two-way ANOVA and the exact *P* values are shown in the figures. Source data are available online for this figure.

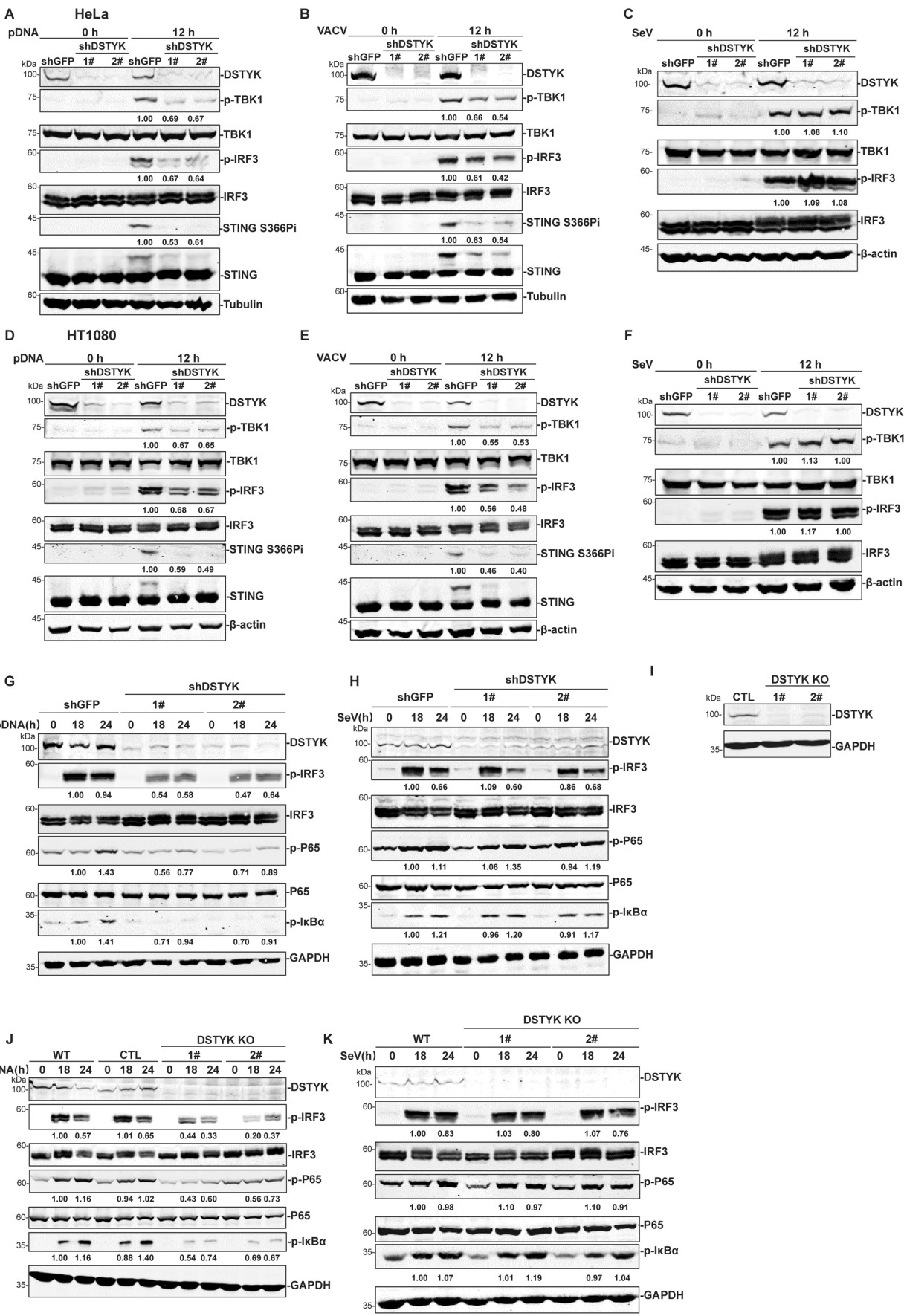

◀ **Figure EV3.  DSTYK deficiency inhibits the cytosolic DNA-triggered phosphorylation of TBK1, IRF3, STING, P65, and IκBα.**

(A–F) *DSTYK* knockdown in HeLa and HT1080 cells inhibits cytosolic DNA-triggered phosphorylation of TBK1, IRF3, and STING at Ser366. *DSTYK*-knockdown HeLa (A–C) and HT1080 (D–F) cells and control cells (shGFP) were stimulated with pDNA transfection (1 μg/mL; A, D), VACV (B, E) and SeV (C, F) for 12 h, and phosphorylation of TBK1, IRF3 and STING at Ser366 was detected by western blotting. (G, H) *DSTYK* knockdown in HeLa cells inhibits cytosolic DNA-triggered phosphorylation of IRF3, P65, and IκBα. *DSTYK*-knockdown HeLa cells and control cells (shGFP) were stimulated with pDNA transfection (1 μg/mL; G) and SeV (H) for the indicated amounts of time, and phosphorylation of IRF3, P65, and IκBα was detected by western blotting. (I) Detection of *DSTYK* knockout by western blotting. (J, K) *DSTYK* knockout in HeLa cells inhibits cytosolic DNA-triggered phosphorylation of IRF3, P65, and IκBα. *DSTYK*-knockout HeLa cells and control cells (WT and CTL) were stimulated with pDNA transfection (1 μg/mL; J) and SeV (K) for the indicated amounts of time, and phosphorylation of IRF3, P65, and IκBα was detected by western blotting. Data are representative of at least three independent experiments. Quantification of the indicated band intensities was performed in ImageJ, and quantification results are labeled below the indicated bands ($n = 3$ technical replicates). Source data are available online for this figure.

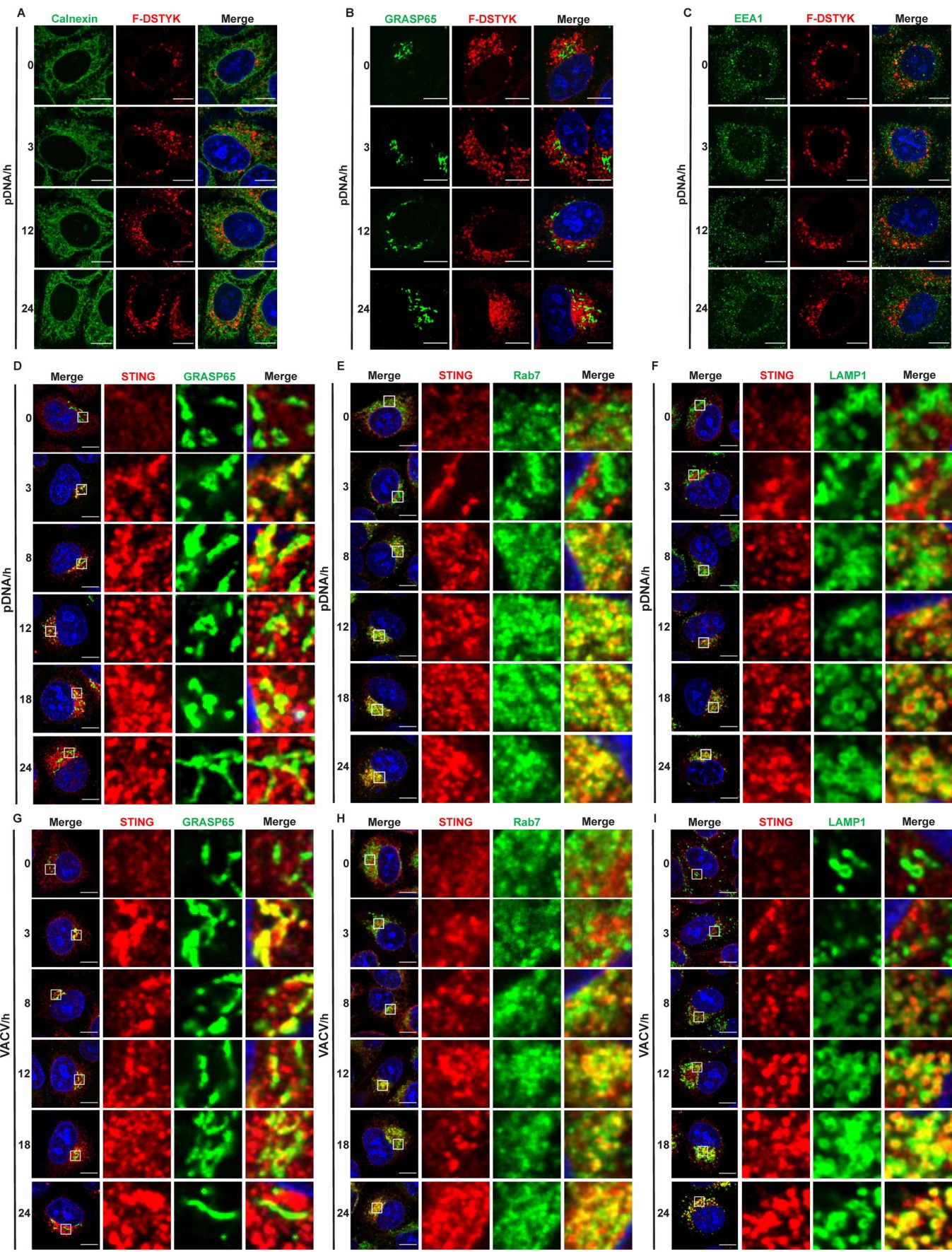

◀ **Figure EV4.  DSTYK does not localize at the ER, Gogi or early endosome, and STING intracellular translocation.**

(**A–C**) DSTYK does not colocalize with the ER, Golgi or early endosome. *DSTYK*-knockout HeLa cells stably expressing DSTYK were stimulated with pDNA transfection (1 μg/mL) for the indicated amounts of time, followed by observation of DSTYK colocalization with calnexin (**A**), GRASP65 (**B**) and EEA1 (**C**) via confocal imaging; scale bars, 10 μm. (**D–F**) STING intracellular translocation induced by pDNA transfection. WT HeLa cells were stimulated with pDNA transfection (1 μg/mL) for the indicated amounts of time, followed by confocal imaging to observe STING colocalization with GRASP65 (**D**), Rab7 (**E**) and LAMP1 (**F**); scale bars, 10 μm. (**G–I**) STING intracellular translocation induced by VACV. WT HeLa cells were stimulated with VACV for the indicated amounts of time, followed by confocal imaging to observe STING colocalization with GRASP65 (**G**), Rab7 (**H**), and LAMP1 (**I**); scale bars, 10 μm. Data are representative of at least three independent experiments.

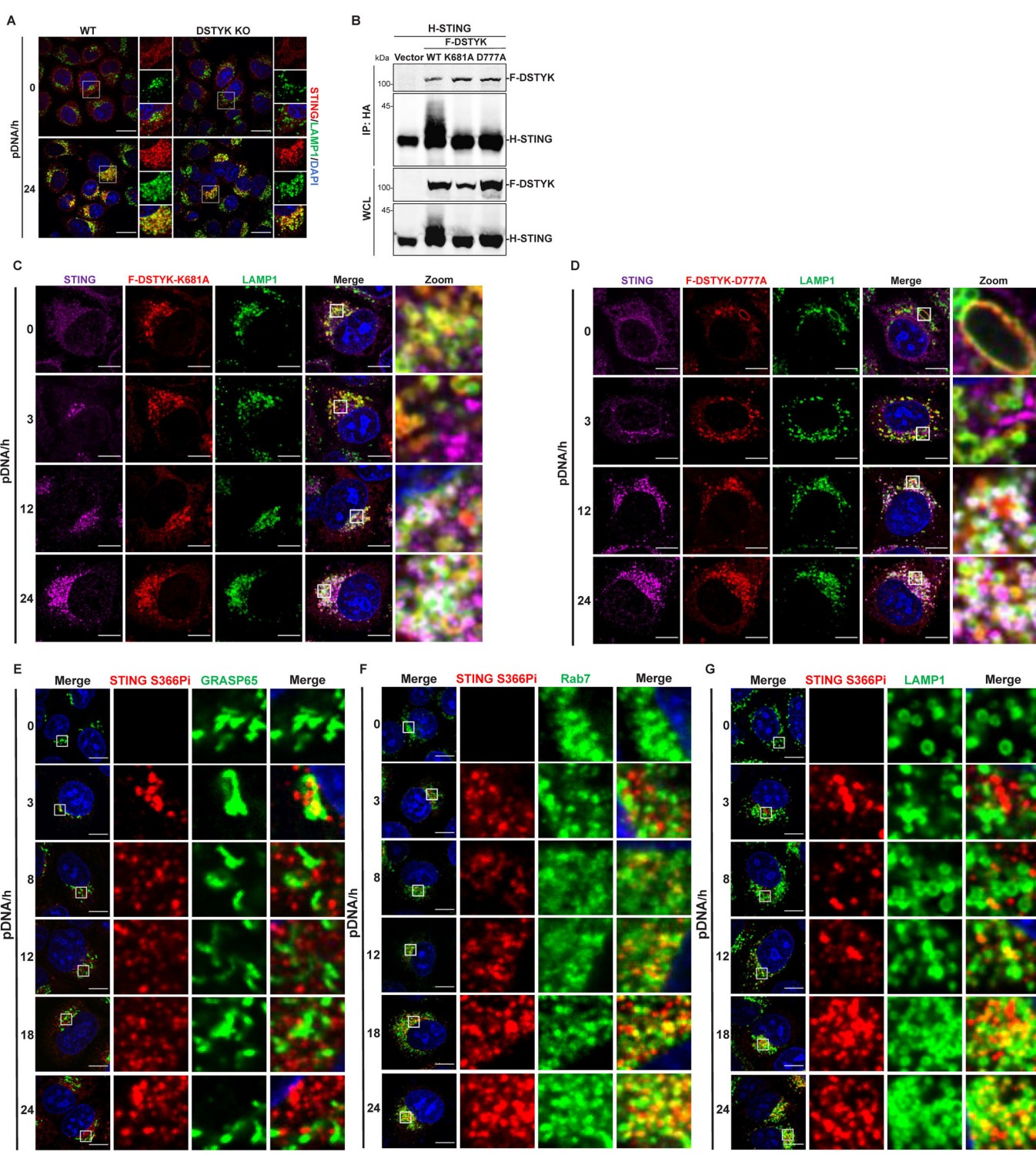

**Figure EV5.   DSTYK does not affect STING translocation, DSTYK kinase-dead mutants can interact with STING, and the translocation of Ser366 phosphorylated STING.**

(**A**) DSTYK does not affect STING colocalization with LAMP1. *DSTYK*-knockout HeLa cells and control cells were stimulated with pDNA transfection (1 μg/mL) for 24 h, followed by confocal imaging to observe the colocalization between STING and LAMP1; scale bars, 20 μm. (**B**) DSTYK kinase-dead mutants interact with STING. HEK293T cells were transfected with the indicated plasmids (5 μg each) for 24 h. Whole-cell lysates (WCL) were examined, and cell lysates were subjected to immunoprecipitation (IP) with anti-HA, followed by immunoblotting (IB) with anti-HA and anti-Flag. (**C**, **D**) DSTYK kinase-dead mutants colocalize with STING. *DSTYK*-knockout HeLa cells stably expressing DSTYK kinase-dead mutants were stimulated with pDNA transfection (1 μg/mL) for the indicated amounts of time, followed by confocal imaging to observe DSTYK colocalization with STING and LAMP1; scale bars, 10 μm. (**E–G**) WT HeLa cells were stimulated with pDNA transfection (1 μg/mL) for the indicated amounts of time, followed by confocal imaging to observe phosphorylated STING colocalization with GRASP65 (**E**), Rab7 (**F**) and LAMP1 (**G**); scale bars, 10 μm. Data are representative of at least three independent experiments.

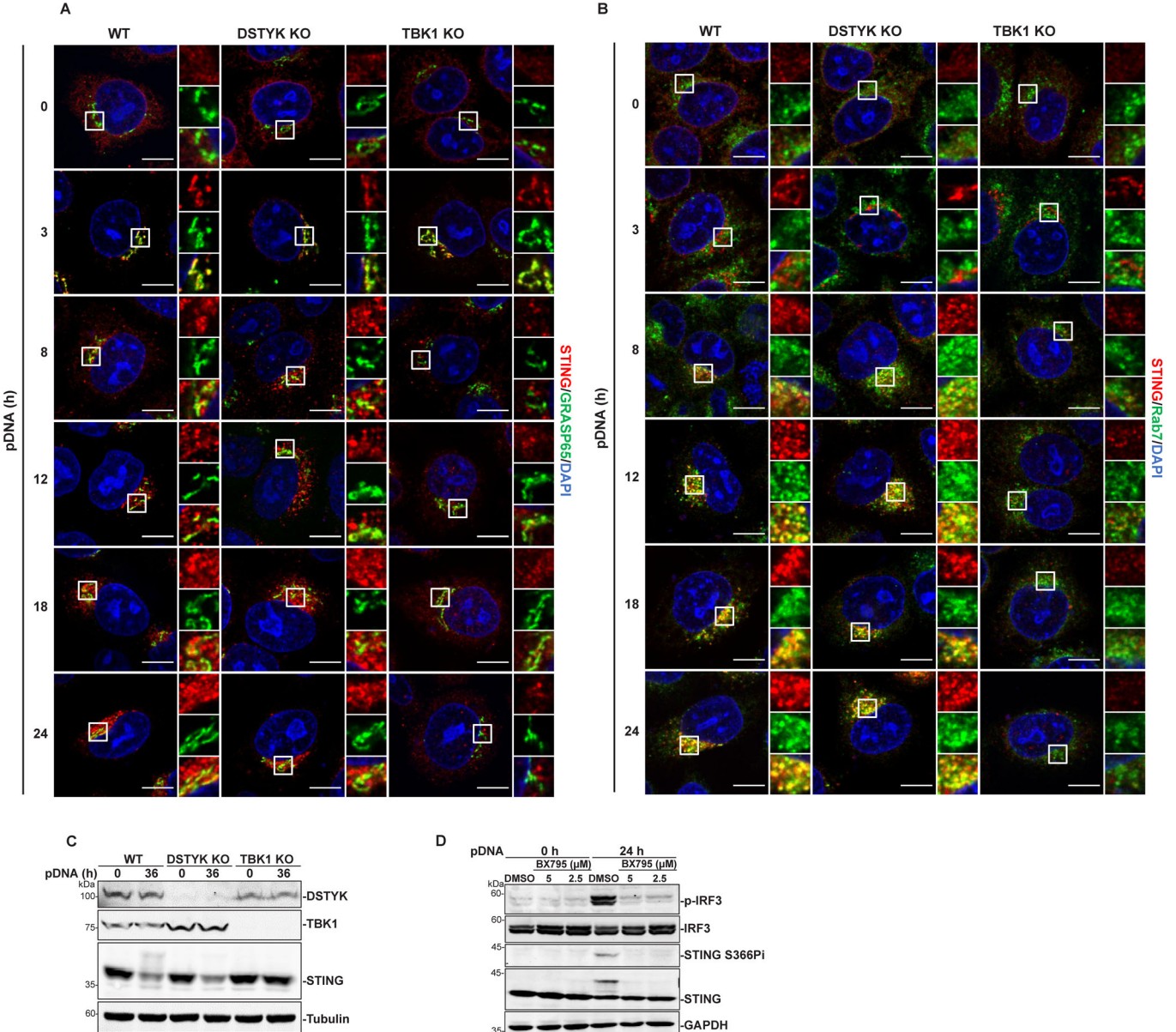

**Figure EV6.   TBK1 knockout inhibits STING post Golgi-trafficking, and BX795 (5 μM and 2.5 μM) inhibits the activation of STING signaling.**

(A, B) TBK1 knockout inhibits STING post-Golgi trafficking. TBK1-knockout HeLa cells, DSTYK-knockout HeLa cells and control cells were stimulated with pDNA (1 μg/mL) for the indicated amounts of time, followed by confocal imaging to observe STING colocalization with GRAPS65 (A) and Rab7 (B); scale bars, 10 μm. (C) TBK1 knockout inhibits STING degradation. TBK1-knockout HeLa cells, DSTYK-knockout HeLa cells and control cells were stimulated with pDNA (1 μg/mL) for 36 h, and STING protein level was detected via western blotting. (D) BX795 inhibits STING signaling. WT HeLa cells were treated with DSMO, BX795 (2.5 μM, 5 μM) for 1 h, then stimulated with pDNA for 24 h, and phosphorylation of IRF3 and STING Ser366 was detected by western blotting. Data are representative of at least three independent experiments.

     

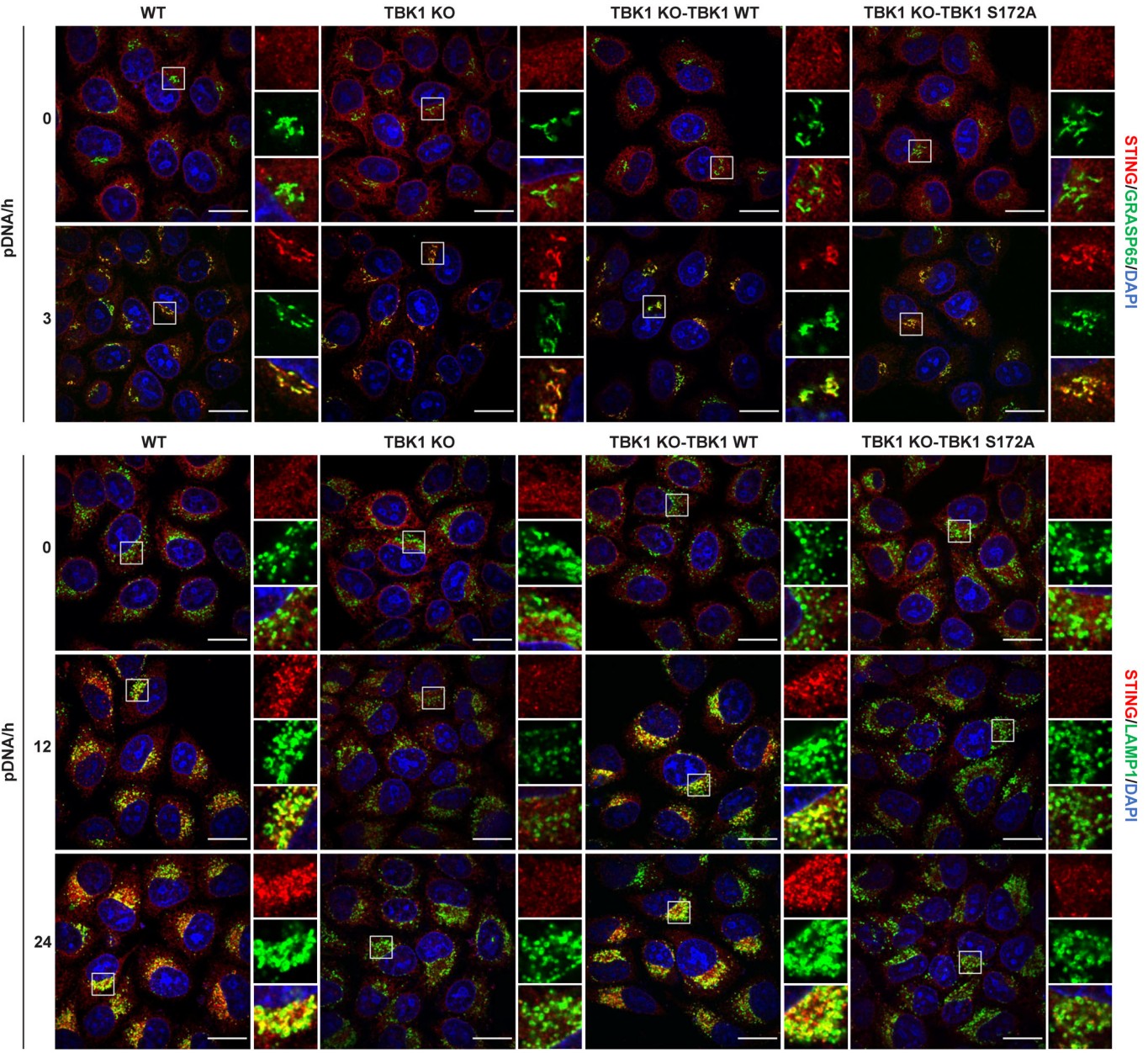

**Figure EV7.  TBK1 controls STING post-Golgi trafficking via its kinase activity.**

*TBK1*-knockout HeLa cells, *TBK1*-knockout HeLa cells stably expressing WT TBK1 (TBK1 KO-TBK1 WT) and kinase-dead mutants (TBK1 KO-TBK1 S172A) and WT HeLa cells were stimulated with pDNA transfection (1 μg/mL) for 3, 12 and 24 h, followed by confocal imaging to observe STING colocalization with GRASP65 and LAMP1, respectively; scale bars, 20 μm. Source data are available online for this figure.

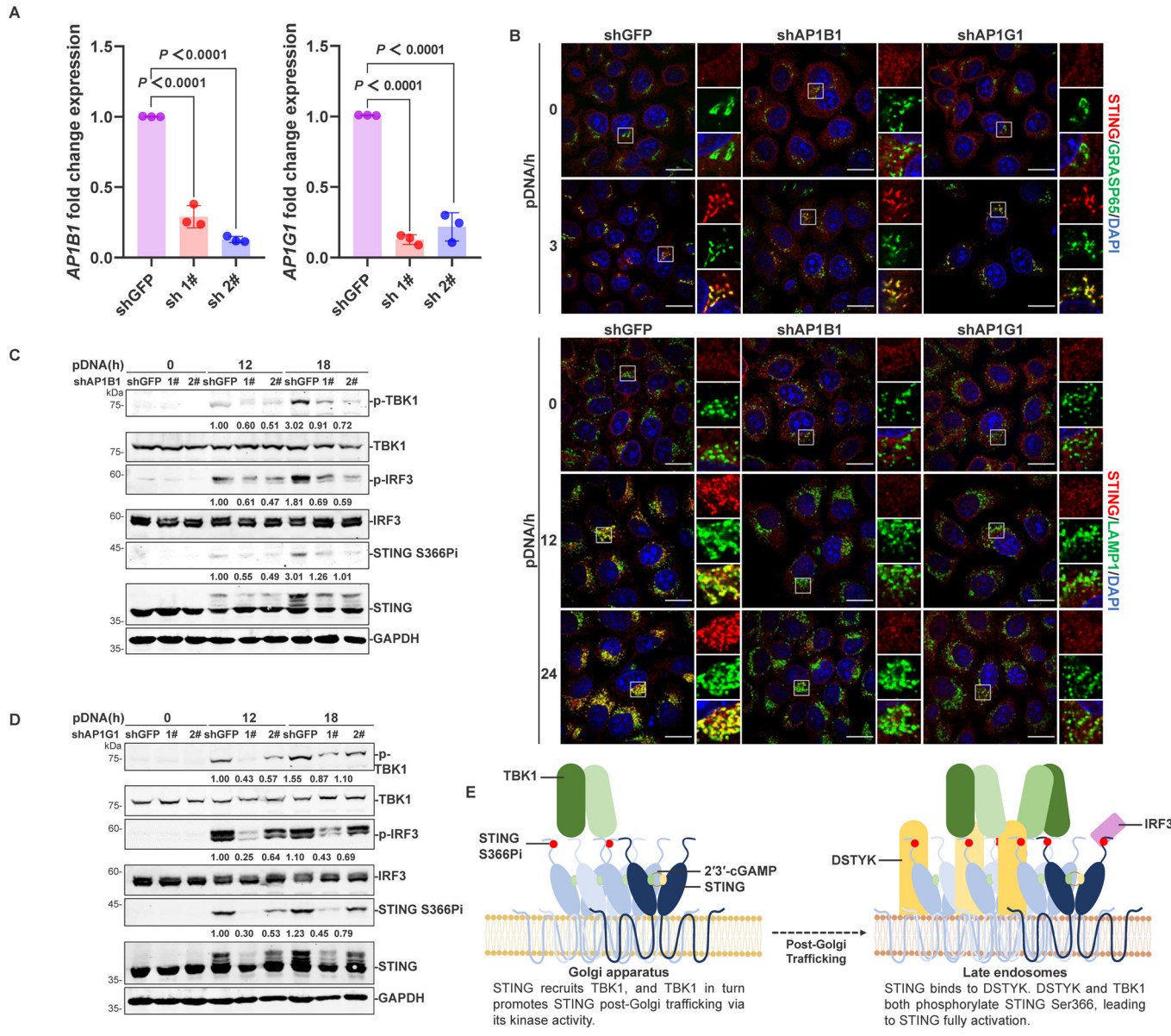

**Figure EV8. Blocking STING post-Golgi trafficking inhibits STING signaling activation, and the schematic diagram.**

(A) Detection of *AP1B1* and *AP1G1* knockdown by qPCR (*n* = 3 biological replicates). (B) AP-1 deficiency inhibit STING colocalization with LAMP1. *AP1*-knockdown HeLa cells and control cells (shGFP) were stimulated with pDNA (1 μg/mL) for 3, 12, and 24 h to observe STING and GRASP65 colocalization as well as STING and LAMP1 colocalization, respectively, by confocal imaging; scale bars, 20 μm. (C, D) AP-1 deficiency inhibit pDNA transfection-induced phosphorylation of TBK1, IRF3, and STING at Ser366. *AP1*-knockdown HeLa cells and control cells (shGFP) were stimulated with pDNA (1 μg/mL) for the indicated lengths of time to observe phosphorylation of TBK1, IRF3, and STING at Ser366 via western blotting. Quantification of the indicated band intensities was performed in ImageJ, and quantification results are labeled below the indicated bands (*n* = 3 technical replicates). (E) The schematic diagram to summarize the working model. Activated STING arrives the Golgi and recruits TBK1, and TBK1 recruitment promotes STING post-Golgi trafficking to late endosomes via its kinase activity. TBK1 and late endosomal-localized DSTYK can both directly phosphorylate late endosomal-localized STING at Ser366, thus further promoting maximized activation of STING signaling. Data are representative of at least three independent experiments. Data in (A) are shown as the mean ± SD and *P* values were analyzed by one-way ANOVA and the exact *P* values are shown in the figures.

