## [Peer Review File · EMBO Reports]

DSTYK phosphorylates STING at late endosomes to promote STING signaling

Hao Dong, Heng Zhang, Pu Song, Yuan Hu, and Dan-Ying Chen

Corresponding author(s): Dan-Ying Chen (dychen@pku.edu.cn), Hao Dong (donghao98@stu.pku.edu.cn)

Review Timeline:

Transfer Date:	29th Jul 24
Editorial Decision:	13th Aug 24
Revision Received:	12th Nov 24
Editorial Decision:	13th Dec 24
Revision Received:	19th Jan 25
Accepted:	29th Jan 25

Transaction Report: This manuscript was transferred to EMBO reports following peer review at The EMBO Journal.

Referee #1:

This manuscript by Dong et al identified a kinase DSTYK that regulates innate immune STING signaling. The authors provided convincing set of data demonstrating that DSTYK knockdown or knockout in cells specifically reduce STING signaling without affecting other innate immune pathways. These results were further confirmed using macrophages and fibroblasts isolated from *Dstyk*^{-/-} mice. The kinase activity of DSTYK is important. The authors also showed that DSTYK phosphorylates the key Ser366 on STING in cells and using recombinant proteins. DSTYK is localized to the endosomes and lysosomes. The authors propose a model in which STING is normally phosphorylated on Ser366 by TBK1 on the Golgi; when it moves to endolysosomes, DSTYK continues to phosphorylate STING to sustain the IFN signaling. This is a plausible model, although some of the mechanistic details are not quite clear.

The major issue is that how could DSTYK acting as a redundant kinase to TBK1 but also depends on TBK1. This is a difficult concept to establish because STING signaling absolutely requires TBK1. The authors argue that TBK1 and DSTYK acts on STING-Ser366 sequentially, but this needs to be more convincingly demonstrated. They claim that STING phosphorylation by TBK1 is required for STING trafficking to endolysosomes where STING will come in contact with DSTYK. I'm not sure this is true; STING trafficking should be normal without Ser366 phosphorylation as shown by the *Sting*-S356A mutant studies. The author should more carefully evaluate whether STING trafficking and degradation kinetics are altered in TBK1 knockout cells, DSTYK knockout cells (in addition to reduced IFN signaling), and better establish two kinases acting in the sequential model.

One idea is to express just the cytoplasmic domain of STING (or target STING to endolysosomes using TMEM192 fusion as in Gentili et al 2023) in TBK1 knockout cells to eliminate the need for trafficking and eliminate the main kinase TBK1, then demonstrate whether DSTYK can phosphorylate STING-CTD and produce IFN signaling. i.e., reconstitute STING/TBK-DKO cells with STING-CTD, which should produce IFN signal when overexpressed, then knockdown DSTYK.

Figure EV5 shows AP-1 knockdown decreased STING-mediated pTBK1 and pIRF3. These data are in conflict with Liu 2022 Nature, where they showed AP-1 knockdown increased both phosphorylations and IFN signaling. It is generally agreed upon that STING Golgi to lysosome trafficking turns off STING signaling. This makes it hard to place how could DSTYK enhance STING signaling at the endolysosomes steps. One possibility is retrograde trafficking from endolysosomes back to the Golgi. Can the authors evaluate the alternative explanation: DSTYK promotes an unknown process that mediates STING trafficking back to the Golgi, which would increase more TBK1-mediated pS366 and IFN signaling.

Minor comments: Figure 1 using knockdown and Figure 2 using KO are redundant for main figures.

Referee #2:

This is an interesting, well-conducted study which describes a role for the kinase DSTYK (RIPK5) in phosphorylating STING and promoting cGAS-STING signalling and resulting type I interferon production. While the data seem sound and the experiments seem well-thought out and executed, I have a major concern about the relevance of these findings considering the cellular models that the authors have used, and would require significantly more evidence, in different cell types, of an endogenous role for DSTYK in phosphorylating STING. My major and minor concerns are listed below:

MAJOR

1. As mentioned, I believe that the models that the authors use throughout the majority of this work (DSTYK overexpression system in cancer cells) are artificial and do not address the question of whether DSTYK regulates STING endogenously in the context of antiviral immunity. As a starting point, the paper does not at any point describe which cell types actually express DSTYK. A quick literature search reveals one publication which claims that DSTYK is not expressed in immune cells (PMID 36263437), which would bring into question the relevance of these findings. Therefore, I would first suggest the authors to measure DSTYK expression in various primary immune cells.

2. Secondly, it is unclear to me why the authors decided to investigate the role of DSTYK in cGAS-STING signalling in the first place? If the idea was to simply investigate the role of DSTYK in innate immunity, then the authors should really test a range of different inflammatory PRR stimuli in the DSTYK KO. This would help the narrative of the story and help the reader to understand the rationale of this study

3. Although Figure 6 does in part address the question of endogenous relevance by showing reduced *Ifnb1* expression in primary DSTYK KO cells, the authors must perform some of the prior mechanistic experiments under similar conditions. For example, a critical piece of evidence supporting a role for DSTYK in the regulation of STING is the immunoprecipitation experiment (Fig. 3C). However, this experiment is highly artificial, performed by overexpressing various tagged plasmids in 'empty' HEK293T cells. The authors should attempt to pull down DSTYK or STING endogenously, preferably in primary immune cells, without the need for overexpression. The authors should also consider repeating some of the other experiments (e.g. the colocalisation experiments) in primary cells

4. Perhaps there is a more unbiased way to detect the interaction between STING and DSTYK, for example by pulling down STING and performing mass spectrometry to detect the various binding partners. This experiment will have been performed previously in other studies, has DSTYK been identified as a STING interactor in other publicly available datasets?

5. While this may be beyond the scope of this work, to conclusively determine a direct interaction between DSTYK and STING, one would cryo-EM to model the interaction, as was performed to model the interaction between TBK1 and STING

MINOR

1. As well as using plasmid DNA transfection as a cGAS-STING agonist, the authors could consider

using a direct STING agonist (e.g. DiABZI) in Figure 1. This would bypass cGAS activity and provide further evidence that any difference resulting from DSTYK KO occurs at the level of STING

2. I would suggest moving figure 1 to the extended data and using the KO data in figure 2 as a new figure 1. It seems redundant to have both KD and KO data in main figures

3. Given that the authors observe an interaction between DSTYK and STING (Fig. 3C), and the TBK1-STING interaction has previously been observed, would one not expect to also see an interaction between DSTYK and TBK1? Or is this because the interactions occur at different time points, or that STING can only bind one protein at a time? A little more insight with regards to this would be helpful

4. In figure EV3B, it appears that STING is pulled down in the vector sample?

5. If I am not mistaken, fig 3C is missing a vector control sample for comparison

6. Could the authors provide some kind of schematic to summarise their findings which diagrammatically describes what is written in lines 317-321?

Dear Danying,

Thank you for the transfer of your research manuscript to EMBO Reports. As my colleague Ieva Gailite from The EMBO Journal informed you, we are interested in publishing your revised manuscript at our journal.

Please revise your manuscript following my suggestions below. Please also provide a point-by-point response to all referee concerns. Your revised manuscript will be treated as a revision at EMBO Reports and will be sent back to the original 2 referees who had evaluated your manuscript at The EMBO Journal.

- Referee 1: the proposed experiment to test the sequential action of TBK1 and DSTYK seems a good suggestion and should be tested to strengthen these data. In addition, please discuss the discrepant findings regarding the AP-1 KD and provide an explanation for the proposed activation of STING by DSTYK on endolysosomes, in light of the prevailing view that STING is turned off during the trafficking from Golgi to lysosomes. Alternative explanations, such as an effect on retrograde trafficking should be discussed but do not have to be tested experimentally.

- Referee 2: please provide data on the expression of DSTYK in immune cells (point 1) and provide further evidence for an endogenous interaction between DSTYK and STING, e.g., by mining public databases (point 3, 4). Point 2 would certainly strengthen the physiological context and might be experimentally addressed. Point 5 is beyond the scope of the current manuscript and should be discussed (or might be tackled using AlphaFold3, but this is not mandatory).

Acceptance of the manuscript will depend on a positive outcome of this second round of review. It is EMBO Reports policy to allow a single round of revision only and acceptance or rejection of the manuscript will therefore depend on the completeness of your responses included in the next, final version of the manuscript.

We realize that it is difficult to revise to a specific deadline. In the interest of protecting the conceptual advance provided by the work, we recommend a revision within 3 months (November 13). Please discuss the revision progress ahead of this time with the editor if you require more time to complete the revisions.

I am also happy to discuss the revision further via e-mail or a video call, if you wish.

*******IMPORTANT NOTE:**

We perform an initial quality control of all revised manuscripts before re-review. Your manuscript will FAIL this control and the handling will be delayed IN CASE the following APPLIES:

- 1) A data availability section providing access to data deposited in public databases is missing. If you have not deposited any data, please add a sentence to the data availability section that explains that.
- 2) Your manuscript contains statistics and error bars based on $n=2$. Please use scatter blots in these cases. No statistics should be calculated if $n=2$.

When submitting your revised manuscript, please carefully review the instructions that follow below. Failure to include requested items will delay the evaluation of your revision. *****

2) individual production quality figure files as .eps, .tif, .jpg (one file per figure).

Please download our Figure Preparation Guidelines (figure preparation pdf) from our Author Guidelines pages <https://www.embopress.org/page/journal/14693178/authorguide> for more info on how to prepare your figures.

3) a .docx formatted letter INCLUDING the reviewers' reports and your detailed point-by-point responses to their comments. As

part of the EMBO Press transparent editorial process, the point-by-point response is part of the Review Process File (RPF), which will be published alongside your paper.

4) a complete author checklist, which you can download from our author guidelines (<<https://www.embopress.org/page/journal/14693178/authorguide>>). Please insert information in the checklist that is also reflected in the manuscript. The completed author checklist will also be part of the RPF.

5) Please note that all corresponding authors are required to supply an ORCID ID for their name upon submission of a revised manuscript (<<https://orcid.org/>>). Please find instructions on how to link your ORCID ID to your account in our manuscript tracking system in our Author guidelines (<<https://www.embopress.org/page/journal/14693178/authorguide#authorshipguidelines>>)

6) We replaced Supplementary Information with Expanded View (EV) Figures and Tables that are collapsible/expandable online. A maximum of 5 EV Figures can be typeset. EV Figures should be cited as 'Figure EV1, Figure EV2' etc... in the text and their respective legends should be included in the main text after the legends of regular figures.

7) Please note that a Data Availability section at the end of Materials and Methods is now mandatory. In case you have no data that requires deposition in a public database, please state so instead of refereeing to the database. See also < <https://www.embopress.org/page/journal/14693178/authorguide#dataavailability>>. Please note that the Data Availability Section is restricted to new primary data that are part of this study.

8) All Materials and Methods need to be described in the main text using our 'Structured Methods' format, which is required for all research articles. According to this format, the Methods section includes a Reagents and Tools Table (listing key reagents, experimental models, software and relevant equipment and including their sources and relevant identifiers) followed by a Methods and Protocols section describing the methods using a step-by-step protocol format. The aim is to facilitate adoption of the methodologies across labs. More information on how to adhere to this format as well as a downloadable template (.docx) for the Reagents and Tools Table can be found in our author guidelines: <https://www.embopress.org/page/journal/14693178/authorguide#structuredmethods>.

An example of a Method paper with Structured Methods can be found here: <https://www.embopress.org/doi/10.15252/msb.20178071>.

9) At EMBO Press we ask authors to provide source data for the main figures. Our source data coordinator will contact you to discuss which figure panels we would need source data for and will also provide you with helpful tips on how to upload and organize the files.

Additional information on source data and instruction on how to label the files are available <<https://www.embopress.org/page/journal/14693178/authorguide#sourcedata>>.

10) The journal requires a statement specifying whether or not authors have competing interests (defined as all potential or actual interests that could be perceived to influence the presentation or interpretation of an article). In case of competing interests, this must be specified in your disclosure statement. Further information: <https://www.embopress.org/competing-interests>

11) Figure legends and data quantification:
The following points must be specified in each figure legend:

- the name of the statistical test used to generate error bars and P values,
 - the number (n) of independent experiments (please specify technical or biological replicates) underlying each data point,
 - the nature of the bars and error bars (s.d., s.e.m.)
- If the data are obtained from n {less than or equal to} 5, show the individual data points in addition to the SD or SEM.
- If the data are obtained from n {less than or equal to} 2, use scatter blots showing the individual data points.

Discussion of statistical methodology can be reported in the materials and methods section, but figure legends should contain a

basic description of n, P and the test applied.

12) Our journal encourages inclusion of *data citations in the reference list* to directly cite datasets that were re-used and obtained from public databases. Data citations in the article text are distinct from normal bibliographical citations and should directly link to the database records from which the data can be accessed. In the main text, data citations are formatted as follows: "Data ref: Smith et al, 2001" or "Data ref: NCBI Sequence Read Archive PRJNA342805, 2017". In the Reference list, data citations must be labeled with "[DATASET]". A data reference must provide the database name, accession number/identifiers and a resolvable link to the landing page from which the data can be accessed at the end of the reference. Further instructions are available at <<https://www.embopress.org/page/journal/14693178/authorguide#referencesformat>>.

13) As part of the EMBO publication's Transparent Editorial Process, EMBO Reports publishes online a Review Process File to accompany accepted manuscripts. This File will be published in conjunction with your paper and will include the referee reports, your point-by-point response and all pertinent correspondence relating to the manuscript.

Kind regards,

Martina

Referee #1:

This manuscript by Dong et al identified a kinase DSTYK that regulates innate immune STING signaling. The authors provided convincing set of data demonstrating that DSTYK knockdown or knockout in cells specifically reduce STING signaling without affecting other innate immune pathways. These results were further confirmed using macrophages and fibroblasts isolated from *Dstyk*^{-/-} mice. The kinase activity of DSTYK is important. The authors also showed that DSTYK phosphorylates the key Ser366 on STING in cells and using recombinant proteins. DSTYK is localized to the endosomes and lysosomes. The authors propose a model in which STING is normally phosphorylated on Ser366 by TBK1 on the Golgi; when it moves to endolysosomes, DSTYK continues to phosphorylate STING to sustain the IFN signaling. This is a plausible model, although some of the mechanistic details are not quite clear.

The major issue is that how could DSTYK acting as a redundant kinase to TBK1 but also depends on TBK1. This is a difficult concept to establish because STING signaling absolutely requires TBK1. The authors argue that TBK1 and DSTYK acts on STING-Ser366 sequentially, but this needs to be more convincingly demonstrated. They claim that STING phosphorylation by TBK1 is required for STING trafficking to endolysosomes where STING will come in contact with DSTYK. I'm not sure this is true; STING trafficking should be normal without Ser366 phosphorylation as shown by the *Sting*-S356A mutant studies. The author should more carefully evaluate whether STING trafficking and degradation kinetics are altered in TBK1 knockout cells, DSTYK knockout cells (in addition to reduced IFN signaling), and better establish two kinases acting in the sequential model.

One idea is to express just the cytoplasmic domain of STING (or target STING to endolysosomes using TMEM192 fusion as in Gentili et al 2023) in TBK1 knockout cells to eliminate the need for trafficking and eliminate the main kinase TBK1, then demonstrate whether DSTYK can phosphorylate STING-CTD and produce IFN signaling. i.e., reconstitute STING/TBK-DKO cells with STING-CTD, which should produce IFN signal when overexpressed, then knockdown DSTYK.

Reply:

We want to explain the statement in our study that DSTYK and TBK1 are not redundant throughout the entire activation process of STING, and do not work on a sequential model. TBK1 controls STING post-Golgi trafficking and regulates the interaction between DSTYK and STING. However, for the phosphorylation on Ser366 of late endosomal-localized STING, the functions of DSTYK and TBK1 can be considered redundant because they can both phosphorylate late endosomal-localized STING. We discuss the two following main points in detail:

1. About STING phosphorylation by DSTYK and TBK1 at late endosomes.

Based on reviewer's recommendations, we used TMEM192-STING-139-379 to mimic sustained late endosomal-localized STING and to further explore the relationship between phosphorylation of late endosomal-localized STING with TBK1 and DSTYK. We stably overexpressed TMEM192-STING-139-379 in WT, *DSTYK* knockout, *TBK1* knockout and double knockout cells (Fig. 5C-F). Stably-expressed TMEM192-STING-139-379 showed phosphorylation of STING Ser366 at resting state, and pDNA stimulation enhanced the phosphorylation. *TBK1* knockout partially inhibited the phosphorylation of TMEM192-STING-139-379 both at resting state and after pDNA stimulation (Fig. 5D,E), which demonstrates that TBK1 can phosphorylate STING Ser366 at late endosomes. Combined with results in Fig. 5A,B that TBK1 colocalizes with STING

and DSTYK at late endosomes and previous report demonstrating that STING gets activated by TBK1 at Golgi (Kojiro et al. PMID: 27324217), we conclude that TBK1 phosphorylates STING Ser366 not only at Golgi but also at late endosomes.

DSTYK knockout showed the same results that *DSTYK* knockout partially inhibited the phosphorylation of TMEM192-STING-139-379 both at resting state and after pDNA stimulation (Fig. 5F). Double knockout of *DSTYK* and *TBK1*, or knockdown *DSTYK* in *TBK1* knockout cells further abolished the phosphorylation of TMEM192-STING-139-379 (Fig. 5D-F). These suggested that TBK1 and DSTYK can both phosphorylate STING Ser366 at late endosomes, and the function of DSTYK in phosphorylating late endosomal-localized STING is independent of TBK1.

Therefore, we suggest that DSTYK and TBK1 do not work on a sequential model, they can both phosphorylates STING Ser366 at late endosomes.

2. About TBK1's function in controlling STING post-Golgi trafficking.

In this study, we demonstrate that TBK1 kinase activity controls STING post-Golgi trafficking to late endosomes, thereby controlling the function of late endosomal-localized DSTYK to STING. We observed STING colocalization with Golgi and late endosomes to measure STING trafficking. And we found that *TBK1* knockout (Fig. 4G), *TBK1* knockout stably-expressing TBK1 kinase-dead mutant (Fig. 4H) and BX795 treatment (Fig. 4I) inhibited STING colocalization with late endosomes without affecting STING colocalization with Golgi.

Based on reviewer's recommendations, we examined STING detailed trafficking kinetics from 0 h to 24 h of DNA stimulation in WT, *DSTYK* knockout and *TBK1* knockout cells via observing STING colocalization with Golgi and late endosome (Fig. 4F and EV5A,B). And the results showed that *DSTYK* knockout did not affect STING trafficking kinetics compared with the control group (Fig. 4F and EV5A,B), but *TBK1* knockout inhibited STING colocalization with Rab7 when STING in WT cells began to gradually colocalize with Rab7 from 8 h to 24 h of DNA stimulation (Fig. 4F and EV5B). And then we examined STING degradation after 36 h of DNA stimulation in WT, *DSTYK* knockout and *TBK1* knockout cells (Fig. EV5C). The results showed that *DSTYK* knockout did not affect STING degradation, but *TBK1* knockout inhibited STING degradation (Fig. EV5C). These results suggested TBK1 controls STING post-Golgi trafficking.

And TBK1 can phosphorylate STING several residues more than Ser366 (Zhao et al. 2019, PMID: 31118511). In this study we do not emphasize that TBK1 controls STING post-Golgi trafficking through the phosphorylation of STING at Ser366 residue; rather, it is possible that TBK1-mediated phosphorylated residues of STING other than Ser366 may control STING post-Golgi trafficking.

In addition to the findings of our study, similar functions of TBK1 have been previously reported by Liu et al (PMID: 36261523), which can also provide a possible mechanistic explanation for our findings. They demonstrate that TBK1 controls STING interaction with AP-1, thereby promoting STING trafficking from Golgi to post-Golgi compartments and STING degradation. they also confirmed that *TBK1* knockout and BX795 treatment inhibited STING lysosomal degradation.

All these demonstrate that TBK1 kinase activity is required for STING trafficking from Golgi to late endosomes. STING fails to reach late endosomes when deleting *TBK1* or inhibiting TBK1 kinase activity, and in this case late-endosomal localized DSTYK cannot interact and phosphorylate STING.

Figure EV5 shows AP-1 knockdown decreased STING-mediated pTBK1 and pIRF3. These data are in conflict with Liu 2022 Nature, where they showed AP-1 knockdown increased both phosphorylation and IFN signaling. It is generally agreed upon that STING Golgi to lysosome trafficking turns off STING signaling. This makes it hard to place how could DSTYK enhance STING signaling at the endolysosomes steps. One possibility is retrograde trafficking from endolysosomes back to the Golgi. Can the authors evaluate the alternative explanation: DSTYK promotes an unknown process that mediates STING trafficking back to the Golgi, which would increase more TBK1-mediated pS366 and IFN signaling.

Reply:

1. Liu et al. 2022 (PMID: 36261523) and Fang et al. 2023 (PMID: 36921576) both demonstrated that AP-1 promotes STING trafficking from Golgi to post-Golgi compartments, but the functions of AP-1 knockdown in STING phosphorylation and IFN signaling are controversial. Liu et al. 2022 showed that AP-1 knockdown increased STING signaling, but Fang et al. 2023 showed that AP-1 knockdown decreased STING signaling. And in this study, we showed similar results with Fang et al 2023 (Fig. EV6A-D).

2. STING trafficking from Golgi to lysosomes contains two physiological processes: translocation from the Golgi to late endosomes and the fusion of late endosomes with lysosomes. Fang et al. 2023 (PMID: 36921576) has demonstrated that STING trafficking from Golgi to late endosomes is vital for the activation of the STING signaling. And Gui et al. (PMID: 30842662) proposed a model for the activation of STING at late endosomes. These has demonstrated that late endosome can act as an important platform for the activation of STING signaling. And in this study, we further suggest that DSTYK can phosphorylate STING at late endosomes to promote the activation of STING signaling.

3. Based on reviewer's recommendations, we examined STING trafficking kinetics from 0 h to 24 h of DNA stimulation in WT and DSTYK knockout cells (Fig. 4F and EV5A,B). As we discussed earlier, DSTYK knockout cells showed similar STING trafficking kinetics with WT cells.

We cannot rule out the possibility that DSTYK promotes an unknown process that mediates STING trafficking back to the Golgi, but the STING S366Pi should be primarily on the Golgi based on this explanation, whereas in this study, we observed that STING S366Pi occurred primarily after STING left Golgi (Fig. 3F and EV4E-G). So, we suggest that DSTYK may not promote STING trafficking back from late endosomes to Golgi.

Minor comments: Figure 1 using knockdown and Figure 2 using KO are redundant for main figures.

Reply:

Based on reviewer's recommendations, we moved Fig.1 to extended data and used Fig.2 as a new Fig.1.

Referee #2:

This is an interesting, well-conducted study which describes a role for the kinase DSTYK (RIPK5) in phosphorylating STING and promoting cGAS-STING signalling and resulting type I interferon production. While the data seem sound and the experiments seem well-thought out and executed, I have a major concern about the relevance of these findings considering the cellular models that the authors have used, and would require significantly more evidence, in different cell types, of an

endogenous role for DSTYK in phosphorylating STING.

My major and minor concerns are listed below:

MAJOR

1. As mentioned, I believe that the models that the authors use throughout the majority of this work (DSTYK overexpression system in cancer cells) are artificial and do not address the question of whether DSTYK regulates STING endogenously in the context of antiviral immunity. As a starting point, the paper does not at any point describe which cell types actually express DSTYK. A quick literature search reveals one publication which claims that DSTYK is not expressed in immune cells (PMID 36263437), which would bring into question the relevance of these findings. Therefore, I would first suggest the authors to measure DSTYK expression in various primary immune cells.

Reply:

First of all, we want to state that there is some controversy regarding which kinase represents RIPK5 as both DSTYK and ANKK1 have been assigned this role (Gregory et al. PMID: 32732131). DSTYK and ANKK1 both share homology in their kinase domains with other RIP family members. The article cited by reviewer (PMID 36263437) only states that ANKK1 is not expressed in primary immune cells and does not detect the expression of DSTYK in primary immune cells.

Almost all types of mammalian cells express type I interferons (Finlay et al. PMID:25614319). And the innate immune response is carried out by various cell types in animals, not limited to immune cells. In addition to HeLa cells, we also used HT1080 and THP1 cells as experimental systems to obtain functional evidence (Fig. EV1-2).

Based on reviewer's recommendations, we analyzed the expression profile of both human and mouse DSTYK in primary immune cells from the ImmGen server as shown below, and the results showed that DSTYK is widely expressed among different types of primary immune cells.

Figure for referees not shown.

The expression profile of human DSTYK in primary immune cells.

Figure for referees not shown.

The expression profile of mouse Dstyk in primary immune cells.

Values for the expression of human and mouse DSTYK mRNA in various immune cells were extracted from Gene Skyline from the Immunological Genome Project.

(ImmGen; <http://rstats.immgen.org/Skyline/skyline.html>).

We also analyzed the expression profile of human DSTYK in common human cell lines from The Human Protein Atlas as shown below, and the results showed that DSTYK is widely expressed in common human cell lines.

Figure for referees not shown.

Expression profile of human DSTYK in human cell lines.

Values for the expression of human DSTYK mRNA in various human cell lines were extracted from The Human Protein Atlas.

(<https://www.proteinatlas.org/ENSG00000133059-DSTYK/cell+line>).

2. Secondly, it is unclear to me why the authors decided to investigate the role of DSTYK in cGAS-STING signalling in the first place? If the idea was to simply investigate the role of DSTYK in innate immunity, then the authors should really test a range of different inflammatory PRR stimuli in the DSTYK KO. This would help the narrative of the story and help the reader to understand the rationale of this study.

Reply:

Based on reviewer's recommendations, we silenced the expression of *DSTYK* in THP1-derived macrophages, and treated cells with different stimuli: Pam2CSK4 (TLR2 ligand), LPS (TLR4 ligand), poly (I:C) (TLR3 ligand), HSV-1 and SeV. *DSTYK* knockdown did not affect the immune pathway activation induced by Pam2csk4, LPS, poly (I:C) or SeV, but only inhibited HSV-1 induced p-TBK1, p-IRF3 and STING S366Pi, which are recognized as markers for the activation of cGAS-STING pathway. And the data above have been added to the Fig. EV1A-E.

3. Although Figure 6 does in part address the question of endogenous relevance by showing reduced *Ifnb1* expression in primary DSTYK KO cells, the authors must perform some of the prior mechanistic experiments under similar conditions. For example, a critical piece of evidence supporting a role for DSTYK in the regulation of STING is the immunoprecipitation experiment (Fig. 3C). However, this experiment is highly artificial, performed by overexpressing various tagged plasmids in 'empty' HEK293T cells. The authors should attempt to pulldown DSTYK or STING endogenously, preferably in primary immune cells, without the need for overexpression. The authors should also consider repeating some of the other experiments (e.g. the colocalisation experiments) in primary cells.

Reply:

Based on reviewer's recommendations, we examined the endogenous interaction between STING and DSTYK. And the endogenous co-IP results showed that DSTYK interacted with STING after DNA stimulation (Fig. 2D).

And we have tried endogenous immunofluorescence experiments under different conditions, but the DSTYK antibody cannot be used for endogenous immunofluorescence detection. Therefore, we ultimately chose to stably express Flag-tagged DSTYK and conducted immunofluorescence detection using Flag-tag antibody as an alternative approach. And we confirmed DSTYK's localization at late endosomal membrane and also its colocalization with STING, which we believe support our findings.

4. Perhaps there is a more unbiased way to detect the interaction between STING and DSTYK, for example by pulling down STING and performing mass spectrometry to detect the various binding partners. This experiment will have been performed previously in other studies, has DSTYK been identified as a STING interactor in other publicly available datasets?

Reply:

Based on reviewer's recommendations, we added the results of endogenous interaction between STING and DSTYK via co-IP as we discussed above (Fig.2D).

And we also searched the public database BioGRID for the interactor of DSTYK and STING. DSTYK is not identified as a direct STING interactor in BioGRID, but DSTYK and STING share some interactors (such as IKEBE), which indicates the potential interaction between STING and

DSTYK.

Besides, in this study, we observed that DSTYK could directly phosphorylate purified STING *in vitro* (Fig. 3E), and we therefore suggest that there is a direct interaction between DSTYK and STING.

5. While this may be beyond the scope of this work, to conclusively determine a direct interaction between DSTYK and STING, one would cryo-EM to model the interaction, as was performed to model the interaction between TBK1 and STING.

Reply:

Determining the cryo-EM structure of STING with DSTYK is definitely a strong method to define the direct interaction between STING with DSTYK, also the exact mechanisms that underlies DSTYK' function. We discussed this point in the manuscript based on reviewer's suggestion (page 8, line 312-314).

MINOR

1. As well as using plasmid DNA transfection as a cGAS-STING agonist, the authors could consider using a direct STING agonist (e.g. DiABZI) in Figure 1. This would bypass cGAS activity and provide further evidence that any difference resulting from DSTYK KO occurs at the level of STING.

Reply:

In Fig.2A and 2B, we used cGAMP, the direct STING ligand, which could bypass cGAS activity, to stimulate DSTYK-deficient cells and control group cells. And we detected impaired p-TBK1, p-IRF3 and STING S366Pi in DSTYK-deficient cells induced by cGAMP stimulation, which were similar with the results of treatment with DNA and DNA virus. These results further demonstrates that DSTYK does not regulate the activation of cGAS, but regulates the activation of STING signaling.

2. I would suggest moving figure 1 to the extended data and using the KO data in figure 2 as a new figure 1. It seems redundant to have both KD and KO data in main figures.

Reply:

Based on reviewer's recommendations, we moved Fig.1 to extended data and used Fig.2 as a new Fig.1.

3. Given that the authors observe an interaction between DSTYK and STING (Fig.3C), and the TBK1-STING interaction has previously been observed, would one not expect to also see an interaction between DSTYK and TBK1? Or is this because the interactions occur at different time points, or that STING can only bind one protein at a time? A little more insight with regards to this would be helpful.

Reply:

Based on reviewer's recommendations, we detected the colocalization among DSTYK, TBK1 and STING after DNA stimulation. And our results confirmed that TBK1, DSTYK and STING showed colocalization after 24 h of DNA stimulation (Fig. 5B). Together with the results that TBK1 showed colocalization with LAMP1 after 24 h of DNA stimulation (Fig. 5A), we suggest that DSTYK, TBK1 and STING can form complexes at late endosomes.

We also confirmed this conclusion by detecting the function of DSTYK and TBK1 in phosphorylation of late endosomal-localized STING. We stably expressed TMEM192-STING-139-379, which were reported to mimic sustained late endosomal-localized STING (Gentili et al. 2023, PMID: 36739287), and detected the phosphorylation of TMEM192-STING-139-379 using site-specific phosphorylated antibody for STING Ser366 (Fig. 5C-F). *TBK1* knockout partially inhibited the phosphorylation of TMEM192-STING-139-379 (Fig. 5D), and DKO (or knockdown *DSTYK* in *TBK1* knockout cells) further abolished the phosphorylation (Fig. 5D,E). These results confirmed that DSTYK and TBK1 can both phosphorylate STING Ser366 at late endosomes.

However, our co-IP results showed that DSTYK did not interact with TBK1 in STING-non-expressing HEK293T cells (Fig. 2C). This indicates that DSTYK and TBK1 may not have a direct interaction in the absence of STING, but they can form complexes with STING. We added relevant information to the discussion (page 9, line 361-364).

Combined with the colocalization among DSTYK, TBK1 and STING, these results showed that DSTYK, TBK1 and STING can form a complex at late endosomes.

4. In figure EV3B, it appears that STING is pulled down in the vector sample?

Reply:

We checked the EV3B (EV4B in revised manuscript) and found that we incorrectly labeled “IP: HA” as “IP: Flag”.

We performed a pull-down assay using HA-STING to determine whether either the wild-type Flag DSTYK or its kinase-dead mutants could be co-precipitated. And the first lane contains HA-STING along with vector control. We have replaced the correct image.

5. If I am not mistaken, fig 3C is missing a vector control sample for comparison.

Reply:

We have replaced the correct image with a vector control group (Fig. 2C in revised manuscript).

6. Could the authors provide some kind of schematic to summarise their findings which diagrammatically describes what is written in lines 317-321?

Reply:

Based on reviewer’s recommendations, we provided the schematic diagram to summarize our model (Fig. EV6E).

Dear Dr. Chen

- Figure 4 seems very data-rich. Might the confocal images not benefit from having more space in a separate figure? This is a suggestion and I leave that up to you, of course.

- Please remove the Reagents and Tools table from the manuscript and upload it as separate file (file type "Reagent Table"). Our typesetters will insert the table later.

- Experimental animals: please provide the reference number for approval in the methods section.

- Statistical analysis in most of the figures is based on technical replicates rather than biological replicates and should therefore be removed, as $N = 1$. You note that these data are representative of 3 independent experiments and I therefore strongly suggest showing the quantification across these three independent repeats, in which case the statistical analysis would be justified.

- If the statistical analysis remains part of the paper, please note that we ask for the definition of the exact p-values and that the p-values in the figure and the legend correspond. It is e.g., not necessary to define ** if this p value is not shown in the figure panel.

As it stands, our data editors noted the following points that would have to be rectified, but as I pointed out above, please refrain from a statistical analysis based on technical replicates

- Please note that the exact p values are not provided in the legends of figures 1a; 3a; 4d, f; 6a, c, e-g; EV 1f, h; EV 6a.

- Please note that in figures 1a; 3a; 4d, f; 6a, c, e-g; EV 1f, h; EV 6a; there is a mismatch between the annotated p values in the figure legend and the annotated p values in the figure file that should be corrected.

- We perform a routine check on .xls quantification files. Could you please double-check the attached color-coded files? The numbers for the three technical replicates for DSTYK KO seem to be the same. It might well reflect the accurate and good reproducibility of the technical replicates, but just to make sure there was not copy-paste error. Similar for the other files. Thank you very much.

- I introduced a few minor changes in the abstract (see below), mainly to avoid the unspecific term "regulates".

- Finally, EMBO Reports papers are accompanied online by

A) a short (1-2 sentences) summary of the findings and their significance,

B) 2-3 bullet points highlighting key results and

C) a schematic summary figure that provides a sketch of the major findings (not a data image).

Please provide the summary figure as a separate file in PNG or JPG format at a size of 550x300-600 pixels (width x height).

Please note that the size is rather small and that text needs to be readable at the final size. Please send us this information along with the revised manuscript.

Kind regards,

Martina Rembold, PhD

Senior Editor

EMBO reports

=====

Referee #1:

I commend the authors for performing experiments which have satisfactorily addressed my concerns. The endogenous IP experiment and the DSTYK/TBK1 colocalization experiments in particular are important new pieces of evidence which greatly add to the story.

My one remaining comment is to potentially include the data concerning DSTYK expression which the authors provided in their reviewer response, or at least include some comment in the text alluding to the DSTYK expression profile across cell types as this may be of interest to readers.

To conclude, I would be happy to accept this manuscript for publication in EMBO Reports.

Referee #2:

The additional experiments demonstrated a role for TBK1 and DSTYK to each individually phosphorylates STING at the endosomal location. This is novel for STING biology and supports the main conclusion of this study. I have no more concerns.

=====

Abstract with minor changes

Stimulator of interferon genes (STING) is essential for innate immune pathway activation in response to pathogenic DNA. Proper activation of STING signaling requires STING translocation and phosphorylation. Here, we show that dual serine/threonine and tyrosine protein kinase (DSTYK) directly phosphorylates STING Ser366 at late endosomes to promote the activation of STING signaling. We find that TBK1 promotes STING post-Golgi trafficking via its kinase activity, thereby enabling the interaction between DSTYK and STING. We also demonstrate that DSTYK and TBK1 can both promote STING phosphorylation at late endosomes. Using an in vivo *Dstyk*-knockout model, we showed that mice deficient in DSTYK demonstrate reduced STING signaling activation and are more susceptible to infection with a DNA virus. Together, we reveal the previously unknown cellular function of DSTYK in phosphorylating STING and our findings provide insights into the mechanism of STING signaling activation at late endosomes.

All editorial and formatting issues were resolved by the authors.

Dr. Dan-Ying Chen
Peking University
School of Life Sciences
Beijing 100871
China

Dear Dr. Chen,

I am very pleased to accept your manuscript for publication in the next available issue of EMBO reports. Thank you for your contribution to our journal.

Yours sincerely,
